# Learning Sparse Visual Representations via Spatial-Semantic Factorization

**Theodore Zhengde Zhao** [1]  **Sid Kiblawi** [1]  **Jianwei Yang** [2 3]  **Naoto Usuyama** [1]  **Reuben Tan** [1]  **Noel C Codella** [1]
**Tristan Naumann** [1]  **Hoifung Poon** [1]  **Mu Wei** [1]

## Abstract

Self-supervised learning (SSL) faces a tension between semantic understanding and image reconstruction. High-level semantic SSL methods such as DINO encourage transformation-invariant global representations for augmentation alignment, while reconstruction-oriented methods such as MAE preserve dense feature grids for spatial grounding but often yield weaker semantic abstractions. We introduce STELLAR, a framework that alleviates this tension by factorizing visual features into a low-rank product of semantic concepts and their spatial distributions. This disentanglement allows us to perform augmentation alignment on semantic tokens while maintaining spatial localization through a localization matrix for reconstruction. We demonstrate that as few as 16 sparse tokens under this factorized form can simultaneously support high-quality reconstruction (2.60 FID) and strong semantic transfer (79.10% ImageNet accuracy). Our results highlight STELLAR as a versatile sparse representation that bridges discriminative and generative vision by separating semantic identity from spatial geometry. Code is available at https://github.com/microsoft/STELLAR.

## 1. Introduction

Learning visual representations has been a central pursuit in computer vision since the advent of deep learning (Bengio et al., 2013). Modern vision models encode raw pixels into latent features powering nearly all downstream applications. Despite advances from early convolutional networks (Lecun et al., 1998) to ResNets (He et al., 2016) and vision transformers (ViTs) (Dosovitskiy et al., 2020), the geometric format of visual representation has remained largely unchanged: a dense 2D grid of high-dimensional features, where each vector is tied to a local patch. This design is intuitive, as it mirrors the grid-like arrangement of pixels.

On the other hand, the field faces a longstanding dilemma: the pursuit of a unified, *holistic* representation that excels at both high-level semantic understanding and low-level reconstruction. While this synthesis has succeeded in natural language processing, where reconstruction tasks like bidirectional masking (BERT (Devlin et al., 2019)) or autoregressive modeling (GPT (Brown et al., 2020)) naturally induce superior semantics, it doesn't directly transfer to the vision domain. Representations learned primarily through image reconstruction (e.g., MAE (He et al., 2022), SimMIM (Xie et al., 2022)) often yield semantics that trail behind contemporary state-of-the-art methods. Consequently, recent self-supervised learning (SSL) approaches have largely diverged into two camps: those prioritizing pixel-level grounding via reconstruction, and those prioritizing rich semantics via joint-embedding invariance (Van Assel et al., 2025).

We argue that this divergence stems from an *Invariance–Equivariance Tension* in conventional visual SSL representations. Faithful reconstruction requires representations to preserve spatially precise information that changes predictably under transformations such as cropping or shifting. In contrast, high-level semantic recognition benefits from invariance to these transformations. Existing joint-embedding methods such as DINO (Caron et al., 2021) encourage such invariance primarily through global image representations. Although their dense patch features can still retain useful spatial structure and transfer well to dense prediction tasks, the representation most optimized for global semantic alignment is not necessarily the one best suited for spatially faithful reconstruction.

In this work, we show that this tension is not an inevitable trade-off, but can be alleviated by moving beyond a purely dense-grid representation toward a *sparse, factorized latent representation*. This representation family provides a unified latent space in which high-fidelity reconstruction and rich semantics can coexist. Our key insight is that the information necessary to describe a scene can be disentangled into two complementary sparse factors:

---

[1]Microsoft [2]xAI [3]Work done at Microsoft. Correspondence to: Theodore Zhengde Zhao <theodorezhao@microsoft.com>, Mu Wei <muwei@microsoft.com>.

*Proceedings of the 43rd International Conference on Machine Learning*, Seoul, South Korea. PMLR 306, 2026. Copyright 2026 by the author(s).

1. **The "What"**: A set of sparse latent tokens representing invariant visual concepts.

2. **The "Where"**: A set of equivariant coefficients representing their spatial locations.

By disentangling these factors through a low-rank matrix factorization form, we enable a "semantic triage": the model is forced to reconstruct the entire image using a highly compressed bottleneck. This encourages the model to ignore stochastically redundant background pixels and focus on semantically-rich object regions. We propose **STELLAR**, a framework that achieves high-quality reconstruction from as few as 16 tokens while encoding fine-grained semantics in a fully self-supervised manner.

Our contributions are summarized as follows:

- **Sparse Representation**: We propose STELLAR, an efficient form of vision modeling that factorizes an image into a handful of sparse tokens by disentangling *what* concepts are present from *where* they are located.

- **SSL Method**: We introduce a training scheme to learn these representations without annotation. By aligning visual concepts across views using optimal transport, we enforce invariance in the "what" factor while adapting the "where" factor, inducing rich semantics.

- **Empirical Observations**: (i) STELLAR achieves a strong state-of-the-art balance between semantic transfer (IN-1K linear acc. 79.10%) and reconstruction quality (FID 2.60). (ii) Sparse factorized training induces fine-grained, region-aware dense features even without explicit dense supervision, yielding strong transfer on downstream dense prediction tasks.

## 2. Related Work

**Self-supervised Learning.** Modern SSL generally falls into two paradigms. *Joint Embedding* (JE) methods, such as MoCo (He et al., 2020) and the DINO family (Caron et al., 2021; Oquab et al., 2023), prioritize global invariance via multi-view alignment, yielding strong semantics. Conversely, *Masked Image Modeling* (MIM), exemplified by MAE (He et al., 2022) and SimMIM (Xie et al., 2022), emphasizes spatial equivariance through pixel reconstruction. Hybrid methods such as iBOT (Zhou et al., 2021), DINOv2 (Oquab et al., 2023), CMAE (Huang et al., 2023), and CAN (Mishra et al., 2022) combine reconstruction-style or patch-level objectives with contrastive/joint-embedding learning, while SODA (Hudson et al., 2024) explores representations for supporting both semantic and generative behavior. STELLAR differs by explicitly factorizing the latent feature map into semantic concept tokens and spa-

tial localization, allowing invariance and equivariance to be modeled by separate factors.

**Sparse Representation.** A growing body of work replaces dense feature maps with compact embeddings. Sparse R-CNN (Sun et al., 2021) and Mask2Former (Cheng et al., 2022) utilize sparse queries for supervised tasks, while BLIP-2 (Li et al., 2023) and TiTok (Yu et al., 2024) employ sparse tokens for vision–language or generative efficiency. SemMAE (Li et al., 2022) utilizes sparse tokens to guide masking using a pretrained teacher. Related object-centric and slot-based models, such as Slot Attention (Locatello et al., 2020), also decompose scenes into a small set of latent slots with spatial assignments, but are typically designed for explicit object discovery or scene decomposition rather than general-purpose SSL transfer. In comparison, STELLAR learns sparse tokens as the **primary latent representation** and learns in a self-supervised manner while preserving both semantic and reconstructive utility.

**Disentanglement & Low-rank Factorization.** The assumption that high-dimensional data lie on low-dimensional manifolds is foundational to dictionary learning (Mairal et al., 2008). In deep learning, low-rank constraints are typically applied to weights for efficiency (e.g., LoRA (Hu et al., 2022)). STELLAR differs by applying *low-rank factorization to the feature map itself*, disentangling "what" (semantic latents $S$) from "where" (spatial assignments $L$).

**The Empirical Dilemma.** Current vision frameworks face a persistent gap: models excelling at pixel-level reconstruction often produce weaker semantic representations (Zhang et al., 2022; Chen et al., 2024), while those achieving top-tier semantics often reduce or abandon pixel-level reconstruction to avoid low-level shortcuts (Assran et al., 2023; Darcet et al., 2025). We demonstrate that by factorizing the latent representation, it is possible to achieve strong performance on both image understanding and reconstruction.

## 3. Preliminaries

Representation learning involves encoding an image $X \in \mathcal{X}$ to latent features $\boldsymbol{Z}(X)$ for downstream tasks. Traditionally, vision representations take a *dense* spatial form:

$$\boldsymbol{Z} \in \mathbb{R}^{n \times d},$$

where $n = h \times w$ denotes the number of patches on a dense grid that partitions the image. Each grid location is represented by a feature vector $\mathbf{z}_i := \boldsymbol{Z}_{i,:} \in \mathbb{R}^d$ for $1 \leq i \leq n$. Most vision architectures also incorporate a global representation $\mathbf{z}_0 \in \mathbb{R}^d$, typically obtained via global pooling or a specialized [CLS] token that undergoes self-attention with patch tokens.

Ideally, we want $\boldsymbol{Z}$ to serve as a *holistic* representation of the image $X$, which retains sufficient information about

the image details, while at the same time possesses rich semantics for downstream tasks. Mathematically, we define such representation as follows:

- **Reconstruction**: There exists a decoder $\mathcal{D}$ such that $\mathcal{D}(\mathbf{Z}(X)) \approx X$. This ensures the representation is spatially and texturally grounded in the physical input.

- **Semantics**: For a downstream task with joint distribution $(X, Y) \sim \mathcal{X} \times \mathcal{Y}$, there exists a simple predictor $f \in \mathcal{F}$ (e.g., a linear layer) such that the expected task loss $\mathbb{E}_{(X,Y)}\big[\mathcal{L}(f(\mathbf{Z}(X)), Y)\big]$ is minimized using frozen features. Typically $Y$ reflects human perception.

Current SSL paradigms face an "*Invariance Paradox*", which is the tension between semantic invariance and spatial equivariance. In order to learn high-level semantics, Joint Embedding (JE) methods (e.g. the DINO family) impose *invariance* to spatial transformations, even across aggressive crops that retain only a small fraction of the original image. On the other hand, reconstruction requires spatial detail, because every pixel shift requires a different set of features for precise reconstruction. This results in representations which are highly *equivariant* to the transformation, i.e. the feature map transforms along with the image transformation.

Let $\mathcal{T}$ be a group of spatial transformations (e.g., translations), and let $t_\theta \in \mathcal{T}$ be parametrized by $\theta$. A representation $\mathbf{Z}(X)$ faces this tension when it is encouraged to satisfy two competing desiderata:

- **Semantic Invariance:** The global readout of the representation should be insensitive to $t_\theta \in \mathcal{T}$:

$$\left\| \frac{\partial}{\partial \theta} \phi(\mathbf{Z}(t_\theta \circ X)) \right\|_F \approx 0.$$

- **Spatial Equivariance:** To allow for high-fidelity reconstruction, the representation must track spatial shifts: $\mathcal{D}(\mathbf{Z}(t_\theta \circ X)) \approx t_\theta \circ X$. Informally, assuming the decoder is locally smooth around $\mathbf{Z}(t_\theta \circ X)$, the chain rule and matrix norm inequalities suggest

$$\left\| \frac{\partial}{\partial \theta} \mathbf{Z}(t_\theta \circ X) \right\|_F \gtrsim \frac{\left\| \frac{\partial(t_\theta \circ X)}{\partial \theta} \right\|_F}{L_\mathcal{D}} > 0,$$

where $L_\mathcal{D}$ denotes a local Lipschitz constant of the decoder around $\mathbf{Z}(t_\theta \circ X)$.

## 4. The STELLAR Framework

### 4.1. Sparse Image Modeling

We now consider an alternative to dense grid-based representations, which describe what appears at each individual

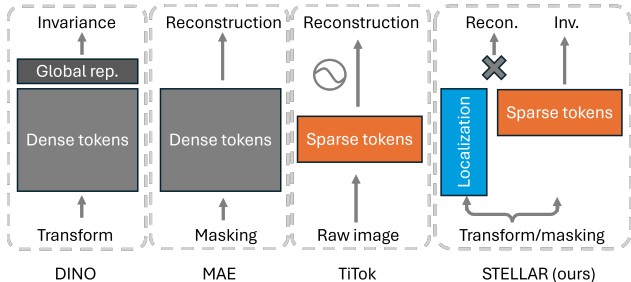

*Figure 1.* Comparison of latent representation learning paradigms. STELLAR learns a sparse factorized latent that separates semantic concepts from spatial localization, enabling semantic alignment and faithful reconstruction within a single representation.

location. We start from the principle that an image depicts the physical world, which can be understood as a collection of objects located in space.

To begin with, we model an image with a compact set of semantic concepts together with their spatial distributions. Let there be $r$ concept embeddings $\boldsymbol{s}_1, \cdots, \boldsymbol{s}_r \in \mathbb{R}^d$, where each $\boldsymbol{s}_j$ captures a distinct semantic concept. The spatial distribution of these concepts is expressed through weights $\boldsymbol{l}_1, \cdots, \boldsymbol{l}_n \in \mathbb{R}^r$, where $n$ is the total number of patches.

By constraining $0 \le \boldsymbol{l}_i \le 1$ and $\mathbf{1}^\top \boldsymbol{l}_i = 1$, each patch is represented as a convex combination of the concept embeddings: $\boldsymbol{v}_i = \sum_{j=1}^r \boldsymbol{l}_{i,j} \boldsymbol{s}_j$. Thus, the set $\boldsymbol{s}_j{}_{j=1}^r$ acts as a basis for constructing local features. In matrix form, the latent representation now takes the form

$$\boldsymbol{Z}(X) = \boldsymbol{L}(X)\boldsymbol{S}(X), \tag{1}$$

where $\boldsymbol{S} = [\boldsymbol{s}_1, \dots, \boldsymbol{s}_r]^\top \in \mathbb{R}^{r \times d}$ is the *semantic matrix*, and $\boldsymbol{L} = [\boldsymbol{l}_1, \dots, \boldsymbol{l}_n]^\top \in \mathbb{R}^{n \times r}$ is the *localization matrix*, with the constraint $0 \le \boldsymbol{L} \le 1, \boldsymbol{L}\mathbf{1}_r = \mathbf{1}_n$. We illustrate the factorized representation in Fig. 1, in contrast to other learning paradigms.

Compared to a canonical dense representation of shape $n \times d$, $\boldsymbol{Z} = \boldsymbol{L}\boldsymbol{S}$ can be considered as a form of low-rank matrix approximation from the sparse representation. While the form resembles the low-rank structure used in convex semi-nonnegative matrix factorization (Ding et al., 2008), $\boldsymbol{S}$ and $\boldsymbol{L}$ are not obtained from any matrix factorization algorithm, but are instead direct outputs of the encoder forward pass, allowing end-to-end training using SSL objectives.

### 4.2. Equivariant Partitioning

The factorized form in equation 1 provides a compact latent interface (storing $(\boldsymbol{L}, \boldsymbol{S})$ requires $r(n + d)$ values instead of $nd$ for a dense representation. With $r = 16$ tokens for a ViT-Base model at $224 \times 224$ resolution, this yields about a 90% reduction in representation size). More importantly, this factorization provides a mechanism to alleviate the

invariance–equivariance tension. The spatial transformation is now partitioned as follows:

$$\underbrace{\frac{\partial \boldsymbol{Z}(t_\theta \circ X)}{\partial \theta}}_{\text{Total Equivariance}} = \underbrace{\left(\frac{\partial \boldsymbol{L}(t_\theta \circ X)}{\partial \theta}\right)\boldsymbol{S}}_{\text{Spatial Equivariance}} + \underbrace{\boldsymbol{L}\left(\frac{\partial \boldsymbol{S}(t_\theta \circ X)}{\partial \theta}\right)}_{\text{Semantic Variance}\approx 0}.$$
$$(2)$$

With spatial and semantic information factorized, spatial equivariance can be primarily absorbed by the localization matrix $\boldsymbol{L}$, while the semantic matrix $\boldsymbol{S}$ can remain comparatively stable across transformations. We require that $\boldsymbol{Z}(X)$ can reconstruct the image by minimizing

$$\mathcal{L}_{recon} = \ell(\mathcal{D}(\boldsymbol{L}(X)\boldsymbol{S}(X)), X). \qquad (3)$$

This *low-rank approximated reconstruction* forces the model to use only sparse tokens $\{\boldsymbol{s}_j\}_{j=1}^r$ to capture sufficient information about the image.

### 4.3. Vision Concept Clustering

To encourage sparse tokens to represent transferable vision concepts, we structure them into $K$ learnable prototypes $\boldsymbol{c}_1, \cdots, \boldsymbol{c}_K \in \mathbb{R}^p$. As shown in Fig. 2, a backbone encoder $\mathcal{E}$ maps a mini-batch of $m$ images into sparse features $\boldsymbol{S}^1, \cdots, \boldsymbol{S}^m$. Each token is projected onto the unit sphere $\mathbb{S}^{p-1}$ via a normalized projector $h : \mathbb{R}^d \to \mathbb{S}^{p-1}$, and its similarity to prototypes $\boldsymbol{C} = [\boldsymbol{c}_1, \cdots, \boldsymbol{c}_K]$ gives logits

$$\lambda_j^i = [\boldsymbol{c}_1 \cdot h(\boldsymbol{s}_j^i), \cdots, \boldsymbol{c}_K \cdot h(\boldsymbol{s}_j^i)], \quad j = 1, \dots, r. \quad (4)$$

Soft assignments over the prototypes are obtained with

$$q_{j,k}^i = \frac{\exp(\lambda_{j,k}^i/\tau)}{\sum_{k'=1}^K \exp(\lambda_{j,k'}^i/\tau)}, \qquad (5)$$

where $\tau$ controls the sharpness. Direct entropy minimization of $q_j^i$ is unstable due to non-convexity and empty clusters. Following (Caron et al., 2020; Darcet et al., 2025), we compute balanced assignments $\tilde{q}_j^i$ from $q_j^i$ using the Sinkhorn-Knopp algorithm with stop-gradient, and minimize

$$\mathcal{L}_{\text{cluster}} = -\frac{1}{mr} \sum_{i=1}^m \sum_{j=1}^r \sum_{k=1}^K \tilde{q}_{j,k}^i \log q_{j,k}^i. \qquad (6)$$

Unlike DINOv2 and SwAV which only use Sinkhorn for balancing teacher targets, we explicitly minimize $\mathcal{L}_{\text{cluster}}$ along with all other objectives.

### 4.4. Set Concepts Alignment

To achieve the semantic invariance in equation 2, we align the sparse tokens $\boldsymbol{s}_1', \dots, \boldsymbol{s}_r'$ obtained from a transformed

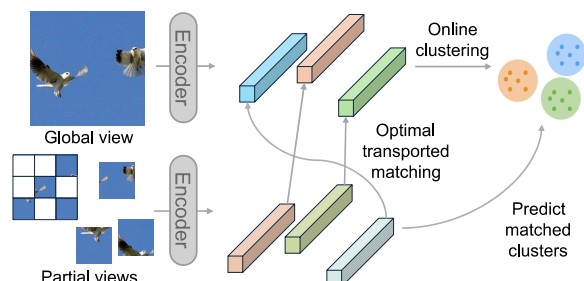

*Figure 2.* Concept clustering and set-alignment workflow. Sparse tokens are assigned to balanced visual prototypes, and tokens from transformed views are matched to those from the global view using optimal transport. The resulting assignments provide view-consistent supervision for learning invariant semantic concepts.

view (e.g. masking or cropping) to the ones from the global view $\boldsymbol{s}_1, \dots, \boldsymbol{s}_r$. However, this set concepts alignment problem is challenging compared to global representation alignment in traditional JE methods, because there is no inherent ordering in the $r$ tokens. To solve the problem, we apply optimal transport with the cost matrix

$$\Theta_{j'j} = \|\boldsymbol{s}_{j'}' - \boldsymbol{s}_j\|_2. \qquad (7)$$

We solve for an assignment matrix $\boldsymbol{P}$ via entropy-regularized optimal transport:

$$\min_{\boldsymbol{P} \geq 0} \quad \sum_{j',j} \boldsymbol{P}_{j'j}\Theta_{j'j} - \epsilon H(\boldsymbol{P}), \qquad (8)$$

$$s.t. \quad \boldsymbol{P}\mathbf{1}_r = \boldsymbol{P}^T\mathbf{1}_r = \frac{1}{r}\mathbf{1}_r, \qquad (9)$$

with $H(\boldsymbol{P}) = -\sum_{j',j} \boldsymbol{P}_{j'j} \log \boldsymbol{P}_{j'j}$. We solve for $\boldsymbol{P}$ using the Sinkhorn algorithm, and define the matching $\sigma(j') := \arg\max_j \boldsymbol{P}_{j'j}$. Compared to bipartite matching algorithms such as Hungarian matching, which are widely used in prior work, this Sinkhorn-based procedure is up to $100\times$ faster in our implementation, as analyzed in the Appendix B.8.

We then compute prototype assignments for the transformed view tokens $q_{j'}' = \text{softmax}(\boldsymbol{C}^T h(\boldsymbol{s}_{j'}')/\tau)$, and minimize the set concept alignment loss

$$\mathcal{L}_{\text{align}} = -\frac{1}{r} \sum_{j'=1}^r \sum_{k=1}^K \tilde{q}_{\sigma(j'),k} \log q_{j',k}'. \qquad (10)$$

Optionally, we use the same framework to cluster and align the CLS token with its own projector and prototypes, similar to previous JE methods. However, we do not use it for reconstruction. We also apply KoLeo regularization (Sablayrolles et al., 2018) on the normalized sparse tokens $\bar{\boldsymbol{s}}_j := \boldsymbol{s}_j/\|\boldsymbol{s}_j\|$ from the same image to encourage concept diversification:

$$\mathcal{L}_{KoLeo} = -\frac{1}{r} \sum_{j=1}^r \log\left(\min_{j' \neq j} \frac{1}{2}\|\bar{\boldsymbol{s}}_j - \bar{\boldsymbol{s}}_{j'}\|_2\right). \qquad (11)$$

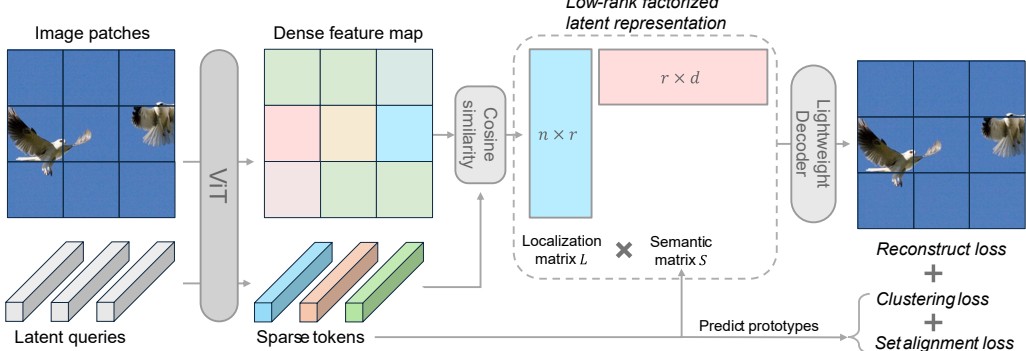

*Figure 3.* Overview of the STELLAR framework. A ViT encoder produces dense patch features and a small set of sparse concept tokens. A localization matrix assigns each patch to these concepts, yielding a low-rank representation $\boldsymbol{Z} = \boldsymbol{LS}$ for reconstruction. Self-supervised clustering and set-alignment losses are applied to the sparse tokens, encouraging them to form transferable visual concepts.

All together, we jointly optimize the following objectives by training the encoder $\mathcal{E}$, decoder $\mathcal{D}$, projector $h$, and prototypes $\boldsymbol{C}$ jointly with the final objective:

$$\min_{\mathcal{E},\mathcal{D},h,C} \quad a_1\mathcal{L}_{\text{recon}} + a_2\mathcal{L}_{\text{cluster}} + a_3\mathcal{L}_{\text{align}}$$
$$+ a_4\mathcal{L}_{\text{cluster-cls}} + a_5\mathcal{L}_{\text{align-cls}} + a_6\mathcal{L}_{\text{KoLeo}}. \quad (12)$$

In summary, we propose a sparse vision representation $(\boldsymbol{S}, \boldsymbol{L}) = \mathcal{E}(X)$ that explicitly disentangles semantic concepts from their spatial distributions, enabling the latent variables to support both pixel-level reconstruction and high-level semantic understanding. We introduce a simple encoder design to obtain these latent variables and SSL objectives to shape them into transferable visual concepts. The full framework overview is illustrated in Fig. 3.

We refer to our framework of learning the spatial-semantic factorized representation $\boldsymbol{Z}(X) = \boldsymbol{L}(X)\boldsymbol{S}(X)$ as **S**parse **T**oken **E**xtraction and **L**ocalization with **L**ow-rank **A**pproximated **R**econstruction (STELLAR).

### 4.5. Model Design

We note that the framework only specifies the latent space, and does not prescribe any specific encoder or decoder architecture. In this work, we adopt a simple design with common modules and model architectures to obtain $\boldsymbol{S}$ and $\boldsymbol{L}$ as described below. We deliberately use standard components to focus on the representation form and SSL objectives, rather than on architectural novelty.

For the encoder part, we use a ViT (Dosovitskiy et al., 2020) as the backbone, and equip it with $r$ learnable latent query vectors, which are passed to the transformer blocks alongside the patch tokens. Processed by the ViT jointly, the latent queries produce sparse tokens $\boldsymbol{S} \in \mathbb{R}^{r \times d}$.

To obtain the localization matrix $\boldsymbol{L} \in \mathbb{R}^{n \times r}$ associated with

the sparse tokens, we use the dense feature map $\boldsymbol{U} \in \mathbb{R}^{n \times d}$ output from the image patches. We project both $\boldsymbol{S}$ and $\boldsymbol{U}$ into a shared embedding space and compute their pairwise cosine similarities, followed by a softmax normalization with temperature $\tau_{\text{spatial}}$ along the sparse-token dimension:

$$\boldsymbol{L} = \text{softmax}\left(\text{cossim}(\boldsymbol{UW}_1, \boldsymbol{SW}_2)/\tau_{\text{spatial}}\right). \quad (13)$$

$\boldsymbol{W}_1$ and $\boldsymbol{W}_2$ are learnable linear projections, and $\tau_{\text{spatial}}$ controls the sharpness of the spatial distribution. We note that this mapping is structurally similar to the attention weights obtained in a single-head cross-attention layer, up to the use of L2 normalization and an explicit temperature parameter. Therefore, the latent representation $\boldsymbol{Z} = \boldsymbol{LS}$ can be viewed as rebuilding a dense feature map for reconstruction by cross-attending to only $r$ sparse concept tokens.

All together, the encoder $\mathcal{E}$ includes ViT transformer blocks, $r$ learnable latent query vectors, and projection layers $\boldsymbol{W}_1, \boldsymbol{W}_2$. The decoder $\mathcal{D}$ is a lightweight 6-layer ViT that reconstructs image patches.

The STELLAR framework can be initialized from a pretrained ViT, such as MAE or DINO, to reshape a strong vision prior into a sparse holistic representation. It can also be trained from random initialization and achieve competitive spatial, semantic, and reconstruction quality. We provide a detailed analysis in the ablation study.

## 5. Experiments

We train STELLAR on ImageNet-1K (Deng et al., 2009) without labels. The encoder is a vanilla ViT (Dosovitskiy et al., 2020) augmented with 8–24 learnable latent queries that produce sparse tokens. A lightweight 6-layer ViT serves as the decoder, predicting MaskGIT-VQGAN tokens (Esser et al., 2021; Chang et al., 2022). When using a pretrained backbone, we ensure that the pretraining was also performed only on ImageNet-1K. We use MAE as the default prior

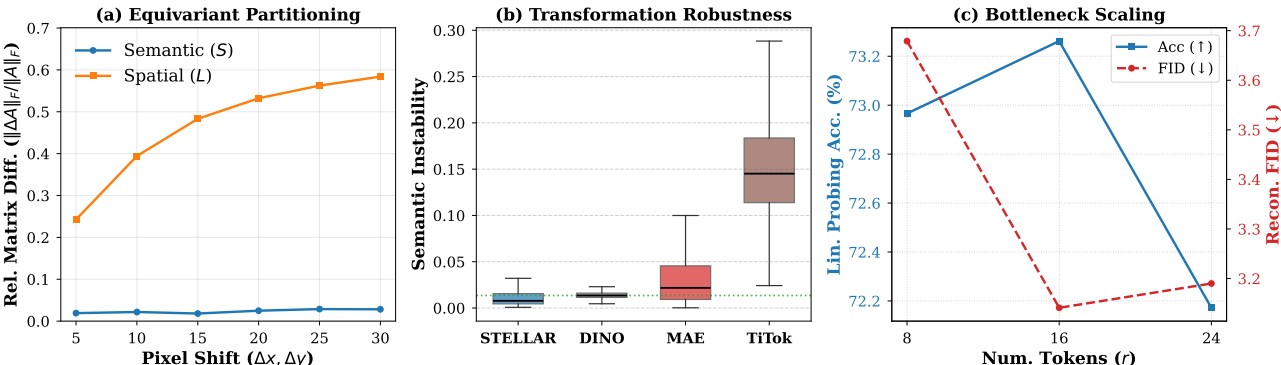

*Figure 4.* Analysis of the STELLAR representation. (a) Relative matrix difference in $L$ and $S$ under controlled pixel shifts in the input image. (b) Cosine distance of latent representations under random crops retaining 50-100% of the image. (c) Impact of the number of sparse tokens $r$ on reconstruction and semantic quality.

and study the effect of different initialization modes in the ablation study. When training from random initialization, we use a momentum-updated encoder to encode the target assignments $\tilde{q}$ in equation 6 and equation 10, following Grill et al. (2020); Caron et al. (2021).

### 5.1. Probing the Factorized Representation

We designed a series of experiments to analyze the factorized representation $Z = LS$ learned by STELLAR.

**Experiment 1: Equivariant Partitioning** We built a controllable and parametrized spatial transformation group to examine the equivariant partitioning in equation 2. Given an image, we take a crop and shift it gradually from 5 to 30 pixels, either horizontally or vertically. We calculated the relative matrix difference $\frac{\|S(t_\theta \circ X) - S(X)\|_F}{\|S(X)\|_F}$ and $\frac{\|L(t_\theta \circ X) - L(X)\|_F}{\|L(X)\|_F}$. Optimal transport in equation 8 is used to match the token ordering. As shown in Fig. 4(a), the semantic matrix $S$ stays nearly invariant, while the spatial localization matrix $L$ changes continuously with the spatial shift, supporting the effectiveness of equivariant partitioning in the factorized representation.

**Experiment 2: Transformation Robustness** Next we compare the semantic stability of STELLAR representation with baseline models under random resized cropping at scale 50-100%. We calculate the cosine distance from the feature of the untransformed image as a measure of semantic instability. For DINO and MAE, we used mean-pooled dense features, and for sparse models STELLAR and TiTok, we used mean-pooled sparse tokens. As shown in Fig. 4(b), the sparse tokens of STELLAR exhibit transformation robustness comparable to DINO. As expected, reconstruction-based models such as MAE and TiTok show higher sensitivity to spatial transformations. In particular, the non-factorized sparse representation from TiTok is

highly unstable, as the model must store both semantic and spatial information in the same space for reconstruction.

**Experiment 3: Effect of Low-rank Bottleneck** The number of sparse tokens $r$ serves as the intrinsic rank of the latent representation. We varied $r$ from 8 to 24 and evaluated reconstruction with FID (Heusel et al., 2017) and semantics with linear probing on mean-pooled sparse tokens. As shown in Fig. 4(c), the linear probing accuracy decreases as $r$ increases, while reconstruction improves, showing a trade-off in the intrinsic rank of the representation. Empirically, $r = 16$ provides a good operating point, achieving both strong semantics and high-quality reconstruction, and we use it as the default for the remaining experiments.

Finally, we visualize the factorized representation in Fig. 5. We show the spatial localization (thresholded by $1/r$) of a sparse token in the image, and the top retrieved semantic concepts from the training dataset.

### 5.2. Evaluating Holistic Representation

Next, we examine the semantic quality and reconstruction potential of representations from different models. For MAE, DINO, and STELLAR, we use a decoder-controlled protocol: the encoder is frozen, and the same lightweight STELLAR decoder is trained on top of the frozen features to reconstruct the image. For STELLAR, we also fine-tune its decoder with the encoder frozen under the same protocol. TiTok is the exception: because its sparse latent is not spatially factorized, we report results using its native decoder, which is a full-sized ViT compared to the 6-layer decoder used in STELLAR.

As shown in Table 1, STELLAR achieves the strongest joint trade-off between semantics and reconstruction among reconstruction-feasible representations. The linear probing and k-NN accuracy of STELLAR surpass all reconstruction-feasible baselines, while trailing the global CLS token from

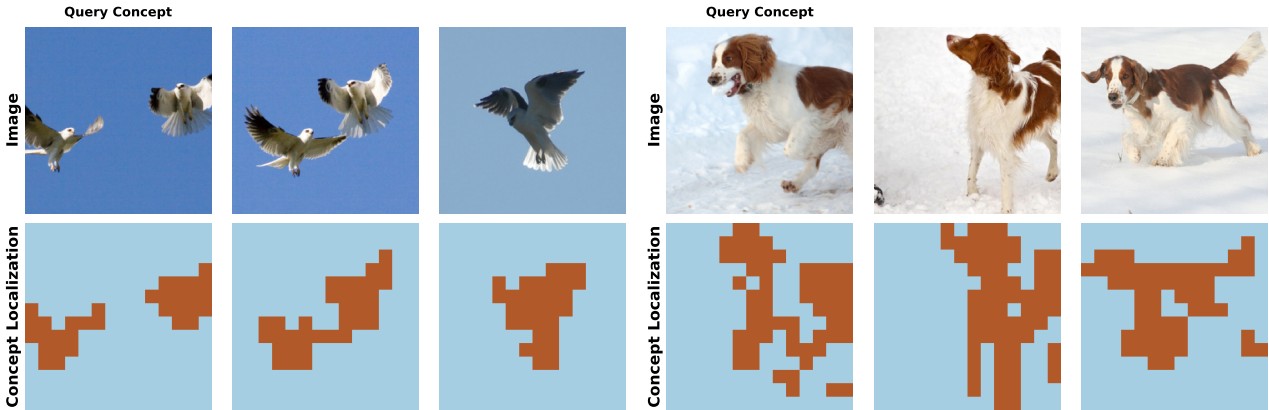

*Figure 5.* Vision concept retrieval with sparse concept tokens. We show both the image and the spatial localization of each concept.

DINO, which is not itself a reconstruction-feasible latent.

*Table 1.* Reconstruction and semantic metrics on IN1K of STELLAR (ours) and baseline models. For reference, we also report semantic metrics of the global representation from DINO and a larger STELLAR model. Best main results are shown in **bold**. Model sizes are ViT-B by default, with larger sizes indicated in parentheses. *: TiTok uses its native larger ViT decoder.

| | | RECONSTRUCTION | | SEMANTICS | |
|---|---|---|---|---|---|
| MODEL | # TKS | FID ↓ | LPIPS ↓ | LIN. | KNN |
| DINO | 1 | - | - | 76.46 | 74.69 |
| DINO | 196 | 3.27 | 0.2121 | 70.31 | 54.41 |
| MAE | 196 | 3.02 | **0.2071** | 66.32 | 25.82 |
| TiTok* | 32 | 2.75 | 0.3281 | 33.42 | 7.30 |
| TiTok* | 64 | 1.99 | 0.2571 | 32.87 | 7.29 |
| OURS | 16 | 3.06 | 0.2077 | **73.26** | **67.25** |
| OURS | 196 | **2.85** | 0.2085 | 72.21 | 64.71 |
| OURS(H) | 16 | 2.60 | 0.1729 | 79.10 | 77.31 |

On reconstruction, STELLAR achieves FID and LPIPS (Zhang et al., 2018) comparable to the dense feature map from MAE, while reducing the latent size by about 90%. Although TiTok achieves lower FID with a much larger decoder, it has substantially worse LPIPS, indicating weaker per-image spatial consistency. In contrast, STELLAR exhibits stronger reconstruction locality with fewer tokens. The full-rank dense feature map $U$ from STELLAR achieves lower reconstruction FID, but with slightly reduced semantic quality. Finally, scaling STELLAR to ViT-H further improves both reconstruction and semantic quality.

### 5.3. Benchmarking Image Understanding

Lastly, we benchmark STELLAR in classical image understanding tasks with linear probing on frozen features, comparing against other ImageNet-pretrained SSL models. We report results for classification on ImageNet-1K (IN1K),

Oxford-IIIT Pet (Pets) (Parkhi et al., 2012), Food-101 (Food) (Bossard et al., 2014), and GlaS (Sirinukunwattana et al., 2016) for cancer grade classification in histopathology. Segmentation benchmarks include ADE20K (Zhou et al., 2017b), Cityscapes (Cordts et al., 2016), and Pascal VOC (Everingham et al., 2010). For broader context, we also include AIM (El-Nouby et al., 2024) and DINOv2 (Oquab et al., 2023), which leverage substantially larger training corpora (100–1000× more images).

As shown in Table 2, the feature map from STELLAR achieves superior performance on ADE20K and Pascal VOC, showing strong fine-grained understanding despite not applying SSL objectives directly to the dense feature map $U$. The results show that sparse factorized modeling implicitly organizes the feature map into semantic regions. Because the image must be reconstructed from a small set of sparse tokens and their spatial localization, each token is encouraged to capture coherent visual content that can be reused across different spatial parts of the scene. This leads to region-aware dense representations. While MAE leads on Cityscapes, STELLAR follows closely with performance comparable to MAE and DINOv2.

On global image understanding tasks, STELLAR achieves the highest IN1K accuracy at large model scale, but smaller variants underperform methods such as DINO that explicitly optimize global representations. Overall, STELLAR outperforms reconstruction-based models and many JE methods, while still trailing the strongest JE models on pure global semantics. Since STELLAR does not model the image as a single global concept, averaging sparse tokens can dilute discriminative information, which is particularly detrimental on object-centric datasets such as Pets and Food. Interestingly, on histopathology images involving complex tissue microenvironments, STELLAR achieves top performance. These results suggest that STELLAR is especially effective for complex, multi-region scenes, while global classification on simple object-centric datasets remains more challenging.

*Table 2.* **Evaluation of Fine-grained and Global Image Understanding.** We evaluate semantic segmentation (mIoU %) and classification accuracy (%) via linear probing on frozen features. For segmentation, we use the dense feature map from the backbone for all models. All numbers are obtained from our own evaluation using official released checkpoints under a consistent protocol. **Bold**: best among ImageNet-trained models. Underline: best within the corresponding architectural class/scale group.

| | | SSL Method | | Segmentation (mIoU) | | | Classification (Acc) | | | |
| Model | Arch. | Target | Objective | ADE20K | CitySc | VOC | IN1K | Pets | Food | GlaS |
|---|---|---|---|---|---|---|---|---|---|---|
| *Semantic-Centric (Joint Embedding / Invariance)* | | | | | | | | | | |
| BYOL | RN-50 | GLOBAL | JE | 18.43 | 18.66 | 63.89 | 70.39 | 82.77 | 64.57 | 95.00 |
| MoCo v3 | ViT-B | GLOBAL | JE | 29.45 | 25.13 | 74.08 | 74.31 | 91.14 | 77.47 | **97.50** |
| DINO | ViT-B | GLOBAL | JE | 26.87 | 26.82 | 79.29 | 76.46 | **93.84** | 79.28 | 95.00 |
| MSN | ViT-B | GLOBAL | JE | 26.66 | 25.39 | 68.59 | 73.65 | 75.91 | 68.93 | 92.50 |
| DenseCL | RN-50 | DENSE | JE | 23.08 | 18.63 | 70.95 | 61.10 | 72.99 | 59.16 | 85.00 |
| Data2Vec | ViT-B | DENSE | LATENT MIM | 22.03 | 23.49 | 61.33 | 54.90 | 26.47 | 34.40 | 73.75 |
| SiameseIM | ViT-B | DENSE | LATENT MIM | 29.24 | 26.52 | 81.38 | 74.97 | 91.61 | 71.01 | 91.25 |
| I-JEPA | ViT-H | DENSE | LATENT MIM | 21.57 | 18.59 | 74.13 | 71.72 | 84.68 | 70.34 | 87.50 |
| iBOT | ViT-B | GL+DE | JE+MIM | 31.78 | 25.69 | 77.06 | 76.40 | 92.40 | 78.08 | 96.25 |
| iBOT | ViT-L | GL+DE | JE+MIM | 33.26 | 26.37 | 77.57 | 78.53 | 92.12 | **81.07** | 96.25 |
| *Image-Centric (Reconstruction)* | | | | | | | | | | |
| BEIT | ViT-B | DENSE | TOKEN MIM | 11.58 | 18.90 | 27.44 | 32.94 | 36.20 | 54.49 | 90.00 |
| BEIT | ViT-L | DENSE | TOKEN MIM | 12.64 | 20.37 | 25.48 | 36.77 | 36.71 | 56.03 | 90.00 |
| SimMIM | Swin-B | DENSE | PIXEL MIM | 12.46 | 17.23 | 35.14 | 24.77 | 27.39 | 40.94 | 77.50 |
| MAE | ViT-B | DENSE | PIXEL MIM | 30.91 | 29.44 | 76.43 | 66.32 | 81.58 | 70.40 | 93.75 |
| MAE | ViT-L | DENSE | PIXEL MIM | 34.36 | 32.53 | 77.79 | 73.09 | 84.30 | 76.22 | 95.00 |
| MAE | ViT-H | DENSE | PIXEL MIM | 36.16 | **35.21** | 78.07 | 75.22 | 84.96 | 78.36 | 95.00 |
| SemMAE | ViT-B | DENSE | PIXEL MIM | 3.52 | 25.48 | 48.33 | 43.84 | 56.99 | 58.90 | 92.50 |
| TiToK-64 | ViT-B | SPARSE | SPARSE REC | – | – | – | 32.87 | 42.06 | 43.68 | **97.50** |
| TiToK-32 | ViT-L | SPARSE | SPARSE REC | – | – | – | 33.42 | 27.83 | 38.83 | 78.75 |
| *Our Method (Sparse Factorized Modeling)* | | | | | | | | | | |
| **STELLAR** | ViT-B | SPARSE | JE+REC | 31.33 | 27.74 | 81.83 | 73.26 | 89.70 | 74.09 | 95.00 |
| **STELLAR** | ViT-L | SPARSE | JE+REC | 34.02 | 31.32 | **85.90** | 76.94 | 92.53 | 74.78 | **97.50** |
| **STELLAR** | ViT-H | SPARSE | JE+REC | **36.66** | 33.30 | 85.66 | **79.10** | 92.53 | 77.43 | 92.50 |
| *Larger Scale Pretraining Beyond ImageNet (Reference Only)* | | | | | | | | | | |
| AIM | 600 M | DENSE | IMAGE AR | 29.00 | 27.04 | 64.55 | 63.78 | 64.68 | 75.19 | 98.75 |
| AIM | 1 B | DENSE | IMAGE AR | 29.59 | 27.05 | 63.90 | 66.86 | 64.21 | 77.96 | 96.25 |
| DINOv2 | ViT-B* | GL+DE | JE+MIM | 40.10 | 34.66 | 89.52 | 82.82 | 95.59 | 91.08 | 98.75 |
| DINOv2 | ViT-L* | GL+DE | JE+MIM | 40.45 | 32.07 | 89.19 | 84.23 | 96.08 | 92.94 | 98.75 |

## 5.4. Ablation Analysis

**Low-rank approximated reconstruction.** As shown in Table 3, removing the low-rank reconstruction objective (A) reduces both global and fine-grained understanding. Since the remaining objectives resemble typical SSL methods, the model still retains reasonable global performance, but fine-grained understanding suffers more. This indicates that low-rank reconstruction encourages sparse tokens to serve as holistic representations covering the entire image.

**Concept clustering.** Eliminating online clustering and set alignment (B) leads to a sharp drop in understanding, highlighting the necessity of structuring sparse tokens into view-invariant concepts. Even when the alignment loss is present (D), removing the clustering loss still leads to a collapse.

**Set alignment.** The training collapsed when training with only reconstruction and clustering (C), underscoring the critical role of set concepts alignment. Additional alignment on the CLS token (E) primarily benefits global classification

but has limited effect on spatial grounding. Finally, KoLeo regularization (F) consistently improves all tasks at a similar level. Interestingly, the absence of either concept clustering or set alignment leads to a sharp drop in performance.

Overall, the sharp drops without concept clustering or set alignment show that the factorized form alone is not sufficient: these objectives are necessary to turn sparse tokens into stable visual concepts. Together, the representation form and the SSL objectives enable STELLAR to learn transferable "what" tokens while preserving spatial grounding through "where" coefficients.

**Effect of foundation prior.** We ablate different initialization modes for STELLAR in Table 4. STELLAR substantially improves the semantic quality of an MAE prior, increasing IN1K linear probing from 66.32 to 73.26, and improves the spatial grounding of a DINO prior, increasing ADE20K from 26.87 to 28.17. When trained from random initialization, STELLAR still reaches MAE-level se-

*Table 3.* **Ablation.** We isolate the impact of each objective on semantic abstraction (IN1K) and spatial grounding (ADE20K), and reconstruction (FID). *Default* denotes the full STELLAR framework. All results are based on ViT-B.

| | Recon. | Cluster | Set Align | CLS Align | KoLeo | rFID ↓ | IN1K ↑ | ADE ↑ |
|---|---|---|---|---|---|---|---|---|
| DEFAULT | ✓ | ✓ | ✓ | ✓ | ✓ | **3.14** | **73.26** | **31.33** |
| *Impact of Individual Components* | | | | | | | | |
| (A) | ✗ | ✓ | ✓ | ✓ | ✓ | — | 72.44 (-0.82) | 29.94 (-1.39) |
| (B) | ✓ | ✗ | ✗ | ✗ | ✓ | 3.21 (+0.07) | 52.07 (-21.19) | 20.46 (-10.87) |
| (C) | ✓ | ✓ | ✗ | ✗ | ✓ | 8.95 (+5.81) | 2.73 (-70.53) | 1.93 (-29.39) |
| (D) | ✓ | ✗ | ✓ | ✓ | ✓ | 3.62 (+0.48) | 42.14 (-31.12) | 18.90 (-12.43) |
| (E) | ✓ | ✓ | ✓ | ✗ | ✓ | 3.26 (+0.12) | 70.79 (-2.47) | 30.20 (-1.12) |
| (F) | ✓ | ✓ | ✓ | ✓ | ✗ | 3.25 (+0.11) | 72.05 (-1.21) | 30.10 (-1.23) |

mantic performance and DINO-level spatial understanding, while maintaining comparable reconstruction quality. These results suggest that the STELLAR objectives can learn a useful factorized representation from scratch, while strong pretrained priors further improve the final representation.

*Table 4.* Evaluating STELLAR trained from different foundational priors. *Base* represents the performance of the original backbone.

| | Recon | Semantic (IN1K) | | Spatial (ADE20K) | |
|---|---|---|---|---|---|
| Prior | FID ↓ | BASE | +STELLAR | BASE | +STELLAR |
| MAE | **3.14** | 66.32 | **73.26** (+6.9) | 30.91 | **31.33** (+0.4) |
| DINO | 3.31 | 76.46 | 73.31 (-3.2) | 26.87 | **28.17** (+1.3) |
| RAND | 3.21 | – | 65.28 | – | 28.10 |

## 6. Discussion and Concluding Remarks

We have shown that the long-standing tension between *semantic abstraction* and *spatial grounding* can be alleviated by changing the form of visual representation. Rather than representing an image only as a dense grid of patch features, STELLAR factorizes the latent space into sparse "What" and "Where" components. This representation allows semantic concepts to remain comparatively stable across views, while spatial localization absorbs the equivariant variation needed for reconstruction.

The core of STELLAR is the principle of *semantic triage*. In a dense representation, every patch receives comparable representational capacity, including background regions or visually redundant details. In contrast, the sparse bottleneck forces the model to explain the image using a small set of reusable visual concepts. To reconstruct the image, these concepts must still remain spatially grounded through the localization matrix. This pressure encourages the model to discover concepts that are both semantically meaningful and useful for explaining the physical image.

The low-rank factorization provides a simple mechanism for separating two complementary aspects of visual information. The *concept tokens* $S$ capture the semantic content of the scene, while the *localization matrix* $L$ describes where these concepts appear. This separation enables semantic alignment losses to act primarily on the "What" component, while reconstruction remains grounded through the "Where" component. In this sense, STELLAR does not remove the need for spatially detailed features, but reorganizes the representation so that semantic invariance and spatial equivariance can be modeled through different factors.

Beyond self-supervised pretraining, sparse factorized representations may provide a useful interface for multimodal models. Modern vision-language systems often pass hundreds of dense visual tokens to a language model, many of which are spatially redundant and difficult to interpret. STELLAR suggests an alternative: a compact set of semantically grounded visual concepts, together with their spatial support. Such a representation could improve visual token efficiency, enable more interpretable cross-modal grounding, and support downstream tasks that require both object-level reasoning and spatial localization.

Several limitations remain. First, while STELLAR can be trained from random initialization, its strongest results are obtained when reshaping a strong pretrained vision prior. We view this as a promising direction for self-supervised representation reshaping or continual pretraining, but future work should further improve from-scratch training. Second, although the sparse factorized latent provides a compact interface, the encoder still computes dense patch features to obtain the localization matrix. Therefore, the main efficiency gain is in representation compression and downstream token processing rather than lower encoder-side compute. Third, the best number of sparse tokens may depend on image complexity, resolution, and downstream use cases. Extending STELLAR to higher-resolution images, video, and open-vocabulary multimodal systems is an important direction for future work.

Overall, our results suggest that sparse factorized tokens are a promising representation format for unifying semantic abstraction, spatial grounding, reconstruction, and efficient multimodal interfacing in self-supervised vision learning.

## Impact Statement

This work advances self-supervised visual representation learning by introducing compact, semantically grounded, and reconstruction-capable image representations. Potential positive impacts include more efficient visual tokenization, improved representation reuse across recognition and reconstruction tasks, and more interpretable visual features for multimodal systems.

As with other general-purpose visual representation learning methods, STELLAR may also be adapted to sensitive applications. These risks are not specific to our method, but improved semantic understanding and reconstruction capability should be deployed with appropriate privacy, data-governance, and application-specific safeguards.

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

# A. Implementation Details

## A.1. STELLAR Training

We train STELLAR with ViT-Base, ViT-Large, and ViT-Huge backbones, each augmented with learnable latent queries, projection layers, prototype heads, and a lightweight 6-layer ViT decoder. Unless otherwise specified, we initialize the ViT backbone from a public MAE checkpoint pretrained on ImageNet-1K, while all newly introduced STELLAR components are randomly initialized. We train STELLAR-B for 150 epochs, STELLAR-L for 100 epochs, and STELLAR-H for 50 epochs. All models are trained on 16 NVIDIA A100-80GB GPUs with a per-GPU batch size of 128, resulting in a total batch size of 2048. We use AdamW (Loshchilov, 2017) with base learning rate $1.5 \times 10^{-4}$ for STELLAR-B and $5 \times 10^{-5}$ for STELLAR-L/H.

For concept clustering, we use $K = 16384$ prototypes for sparse tokens and another $K = 16384$ prototypes for the CLS token. The projector $h$ is a 2-layer MLP before the prototype layer. Balanced prototype assignments are computed with 3 Sinkhorn-Knopp iterations using stop-gradient targets. The spatial localization matrix $L$ is computed from sparse-dense cosine similarities with softmax temperature $\tau_{\text{spatial}} = 0.06$.

For data augmentation, each image is transformed into multiple views. We use 6–8 randomly masked views for sparse-token set alignment, and an additional 6–8 local crops for CLS-token alignment. Global views are sampled with random-resized-crop scale between $36\%$ and $100\%$, while local views use scale between $6\%$ and $36\%$. We also apply standard photometric augmentations, including color jittering, grayscale conversion, and Gaussian blurring.

For the random-initialization ablation, we train STELLAR from scratch and use an exponential moving average (EMA) momentum encoder during the warm-up stage to produce target prototype assignments in equation 6 and equation 10. The EMA encoder includes the ViT backbone, latent queries, projection layers, and prototype heads, and is updated with momentum 0.996. The momentum encoder processes a global view of the image to produce target assignments for both clustering and set alignment. We use masking ratio 0.6 during the EMA warm-up stage and 0.8 during standard training. The random-initialization model is trained for 150 epochs of EMA warm-up followed by 75 epochs of standard training.

For the loss weights in equation 12, we set $a_1 = 0.05$ for the reconstruction loss, which is implemented as cross-entropy over predicted VQGAN codebook indices. We set $a_2 = a_3 = a_4 = a_5 = 1$ for the sparse-token clustering, sparse-token set alignment, CLS-token clustering, and CLS-token alignment losses. The sparse-token set alignment loss is computed only on randomly masked views, while the CLS-token alignment loss is computed on both masked views and local crops, with the two contributions averaged separately for each view type and then summed. We set the KoLeo regularization weight to $a_6 = 0.1$. KoLeo regularization is applied to both global and masked views. The losses are averaged separately for each view type and then summed.

## A.2. Evaluation Protocol

For STELLAR and all baseline models, we evaluate frozen features using linear probing. For classification tasks, we apply layer normalization followed by a single linear classifier. For segmentation tasks, we apply batch normalization followed by a linear classifier at each patch location. For every benchmark, we reserve $10\%$ of the training set for validation and tune hyperparameters on this split. We sweep learning rates in $\{1, 2, 5\} \times \{10^{-5}, 10^{-4}, 10^{-3}\}$ and $10^{-2}$, and batch sizes in $\{64, 128, 256, 512, 1024, 2048, 4096, 8192\}$.

Since different SSL methods optimize different parts of the representation, we evaluate classification using the feature type on which the corresponding SSL objective is primarily applied. For example, we use the global CLS token for DINO and dense patch tokens for MAE. For STELLAR, we use mean-pooled sparse tokens by default. For segmentation, we use the dense feature map from the backbone for all methods, ensuring a consistent dense-prediction protocol across models.

We further evaluate how linear probing accuracy changes with token choice. As shown in Table 5, the features directly optimized by the SSL objective usually provide the best linear probing performance. DINO relies strongly on the global token, while MAE is slightly better with dense features. iBOT performs best with the global token despite also using masked image modeling. In contrast, STELLAR achieves strong performance from sparse tokens while its dense features remain competitive, suggesting that sparse factorized training also shapes the dense backbone representation.

*Table 5.* ImageNet-1K linear probing accuracy (%) using different feature readouts. We mark in **bold** the features on which the corresponding SSL objective is primarily applied, as well as the best accuracy for each method.

| | DINO | | MAE | | iBOT | | | STELLAR (ours) | |
|---|---|---|---|---|---|---|---|---|---|
| Feature | **Global** | Dense | Global | **Dense** | Global | Dense | **Gl.+De.** | **Sparse** | Dense |
| Lin. Acc. | **76.46** | 70.31 | 65.61 | **66.32** | **76.40** | 71.44 | 71.58 | **73.26** | 72.21 |

Table 6 summarizes the baseline models used in our evaluation and categorizes them by their self-supervised learning objective, the prediction space, and the type of tokens on which the SSL objective is primarily applied. This taxonomy helps clarify the comparison in Table 2: different methods optimize different parts of the representation, e.g., global tokens for joint-embedding methods, dense patch tokens for reconstruction-based methods, and sparse tokens for STELLAR.

*Table 6.* List of baseline models and SSL method type.

| **Model** | Reference | Method | SSL space | SSL tokens |
|---|---|---|---|---|
| BYOL | (Grill et al., 2020) | augmentation alignment | latent | global |
| MoCo v3 | (Chen et al., 2021) | contrastive learning | latent | global |
| DINO | (Caron et al., 2021) | augmentation alignment | latent | global |
| MSN | (Assran et al., 2022) | masked alignment | latent | global |
| DenseCL | (Wang et al., 2021) | contrastive learning | latent | dense |
| Data2Vec | (Baevski et al., 2022) | latent MIM | latent | dense |
| SiameseIM | (Tao et al., 2023) | latent MIM | latent | dense |
| IJEPA | (Assran et al., 2023) | latent MIM | latent | dense |
| iBOT | (Zhou et al., 2021) | align + latent MIM | latent | global+dense |
| BEIT | (Bao et al., 2021) | token MIM | image | dense |
| SimMIM | (Xie et al., 2022) | pixel MIM | image | dense |
| MAE | (He et al., 2022) | pixel MIM | image | dense |
| SemMAE | (Li et al., 2022) | pixel MIM | image | dense |
| TiTok | (Yu et al., 2024) | reconstruction + clustering | image | sparse |
| AIM | (El-Nouby et al., 2024) | autoregressive | image | dense |
| DINOv2 | (Oquab et al., 2023) | align + latent MIM | latent | global+dense |

# B. Additional Results

## B.1. Additional Reconstruction Metrics

In addition to FID and LPIPS reported in the main text, we further evaluate per-image reconstruction fidelity using PSNR and SSIM. These metrics provide a complementary view of whether the sparse latent preserves image-specific spatial structure, rather than only producing distributionally plausible reconstructions. As shown in Table 7, STELLAR's low-rank representation achieves PSNR and SSIM comparable to or slightly better than MAE and DINO under the same decoder-controlled protocol. In contrast, although TiTok achieves lower FID with its native larger decoder, it obtains substantially worse LPIPS, PSNR, and SSIM, suggesting weaker per-image fidelity. These results support the conclusion that STELLAR preserves spatial locality well despite using only 16 sparse tokens.

*Table 7.* **Additional reconstruction metrics on ImageNet-1K.** We report PSNR and SSIM in addition to FID and LPIPS. For DINO, MAE, and STELLAR, reconstruction is evaluated under the same decoder-controlled protocol using frozen encoder features and the lightweight STELLAR decoder. TiTok uses its native larger ViT decoder.

| MODEL | # TKS | FID ↓ | LPIPS ↓ | PSNR ↑ | SSIM ↑ | LIN. | KNN |
|---|---|---|---|---|---|---|---|
| | | | RECONSTRUCTION | | | SEMANTICS | |
| DINO | 1 | - | - | - | - | 76.46 | 74.69 |
| DINO | 196 | 3.27 | 0.2121 | 16.36 | 0.3900 | 70.31 | 54.41 |
| MAE | 196 | 3.02 | **0.2071** | 16.54 | 0.3979 | 66.32 | 25.82 |
| TITOK* | 32 | 2.75 | 0.3281 | 14.97 | 0.3311 | 33.42 | 7.30 |
| TITOK* | 64 | 1.99 | 0.2571 | 16.01 | 0.3721 | 32.87 | 7.29 |
| OURS | 16 | 3.06 | 0.2077 | 16.57 | 0.3970 | **73.26** | **67.25** |
| OURS | 196 | **2.85** | 0.2085 | **16.58** | **0.3988** | 72.21 | 64.71 |
| OURS(H) | 16 | 2.60 | 0.1729 | 17.91 | 0.4682 | 79.10 | 77.31 |

## B.2. Pooling Strategy for Sparse Tokens

In the main text, we use mean-pooled sparse tokens for linear probing. To test whether the observed trend over $r$ is an artifact of mean pooling, we additionally evaluate attention pooling and max pooling. As shown in Table 8, $r = 16$ remains the best operating point across pooling strategies, while max pooling performs worse. This suggests that increasing $r$ weakens the semantic bottleneck rather than simply changing the behavior of mean pooling.

*Table 8.* **Effect of pooling strategy on ImageNet-1K linear probing.** We evaluate sparse-token representations with different numbers of tokens $r$ using attention pooling, mean pooling, and max pooling. The trend remains stable across pooling strategies, suggesting that the decrease in accuracy for larger $r$ is not merely a mean-pooling artifact.

| $r$ | ATTENTION POOLING | MEAN POOLING | MAX POOLING |
|---|---|---|---|
| 8 | 72.86 | 72.97 | 72.33 |
| 16 | 73.05 | 73.26 | 72.17 |
| 24 | 72.15 | 72.17 | 70.49 |

## B.3. Sensitivity to Prototype Count

Table 9 studies the effect of the prototype vocabulary size $K$. Performance is stable for $K \geq 4096$, while $K = 1024$ leads to lower ImageNet and ADE20K performance. We also observed less stable clustering and set-alignment losses at $K = 1024$. This suggests that a sufficiently large prototype vocabulary is helpful for modeling fine-grained visual concepts such as objects, parts, materials, textures, and scene context.

## B.4. Effect of Pretraining Data

We pretrained separate STELLAR versions on ImageNet-1K, Places365 (Zhou et al., 2017a) and compared their linear probing performance in Table 10.

*Table 9.* **Sensitivity to the number of visual prototypes** $K$**.** We evaluate ImageNet-1K linear probing and ADE20K segmentation for different prototype vocabulary sizes. Performance is stable once $K$ is sufficiently large, while $K = 1024$ leads to weaker performance.

| $K$ | IN1K ↑ | ADE20K ↑ |
|---|---|---|
| 1024 | 70.79 | 30.21 |
| 4096 | 73.12 | 30.64 |
| 8192 | 73.31 | 30.98 |
| 16384 | 73.26 | 31.33 |

*Table 10.* Effect of pretraining data.

| | linear probing acc. | |
|---|---|---|
| Pretraining data | ImageNet-1K | Places365 |
| ImageNet-1K | 76.94 | 49.25 |
| Places365 | 66.08 | 51.98 |

## B.5. Dense Prediction from the Low-rank Representation

Table 11 compares segmentation probes trained directly on the low-rank representation $LS$ and on the dense feature map. The low-rank representation retains non-trivial spatial information, especially on Pascal VOC, but it does not fully replace dense features for segmentation. Thus, the dense prediction results in the main text should be interpreted as evidence that sparse factorized training shapes the backbone feature map into a more region-aware representation, rather than as evidence that $LS$ alone is sufficient for all dense prediction tasks.

*Table 11.* **Segmentation probing with low-rank and dense representations.** We compare linear segmentation probes trained on the low-rank reconstructed representation $LS$ and on the dense feature map from the backbone. The low-rank representation retains non-trivial spatial signal, but dense features remain substantially stronger for dense prediction.

| REPRESENTATION | ADE20K ↑ | PASCAL VOC ↑ | CITYSCAPES ↑ |
|---|---|---|---|
| LOW-RANK $LS$ | 17.58 | 70.00 | 13.60 |
| DENSE FEATURE MAP | 31.33 | 81.83 | 27.74 |

## B.6. Concept Alignment with Language

Inspired by Zhang et al. (2025), we use frozen STELLAR features and align them with the text tower of CLIP (Radford et al., 2021) using a single attention-pooled probing layer. The evaluation on vision-language tasks, together with comparisons to baseline models, is shown in Table 12. This experiment is intended as a lightweight probing study rather than a fully optimized vision-language training setup.

## B.7. Finetuning

We finetune STELLAR on ImageNet-1K classification and ADE20K segmentation and compare it with baseline models. We use the same train/validation splits and hyperparameter selection protocol as in Sec. A.2, but unfreeze the backbone and finetune all model parameters for 75 epochs. All models use ViT-B backbones. As shown in Table 13, STELLAR achieves consistent gains across tasks and performs close to the strongest baseline, iBOT.

## B.8. Efficiency Analysis

To analyze the efficiency of STELLAR, we measure the processing time of the main components on a single A100 GPU at different batch sizes. Encoding the main global view takes most of the processing time, followed by encoding the masked views (8 views at 80% masking ratio) and decoding to the original image. The Sinkhorn-Knopp algorithm used for clustering and the Sinkhorn algorithm used for optimal transport matching account for a much smaller fraction of the total runtime, and their processing time remains nearly constant as batch size increases.

We compare our Sinkhorn-based matching procedure with Hungarian matching, which is commonly used in prior work

*Table 12.* Language alignment evaluation.

| | IN-1K 0-shot | | MS COCO | | Winoground | | MMVP |
|---|---|---|---|---|---|---|---|
| | @1 | @5 | T2I | I2T | Text | Image | Avg. |
| MAE | 23.18 | 50.43 | 11.28 | 13.46 | 20.75 | 9.00 | 19.26 |
| iBOT | 50.01 | 80.43 | 20.79 | 29.38 | 24.75 | 12.00 | 18.52 |
| STELLAR | 51.53 | 80.04 | 17.94 | 22.34 | 26.25 | 8.25 | 19.26 |
| CLIP | 72.7 | - | 43.0 | 59.7 | 30.5 | 11.5 | 20.0 |

*Table 13.* Finetuning performance in ImageNet-1K classification accuracy and ADE20K segmentation mIOU (%). We show in parentheses the gain over the respective linear probing results.

| Model | ImageNet-1K Acc. | ADE20K mIoU |
|---|---|---|
| DINO | 79.58 (+3.12) | 39.22 (+12.35) |
| MAE | 77.75 (+11.43) | 40.33 (+9.42) |
| iBOT | 80.72 (+9.14) | 42.76 (+10.97) |
| STELLAR | 80.05 (+6.78) | 41.98 (+10.65) |

such as Sparse R-CNN (Sun et al., 2021), DETR (Carion et al., 2020), and MaskFormer (Cheng et al., 2021). Since exact Hungarian matching is less amenable to GPU parallelization, its computational cost increases approximately linearly with batch size. At batch size 64, Hungarian matching already costs more than the encoder forward pass, while our Sinkhorn-based matching is over $100\times$ faster. We therefore add a small entropy regularization term to the bipartite matching objective, enabling efficient GPU-parallel matching with Sinkhorn iterations.

*Table 14.* Processing time (s) of the main components in STELLAR on one A100 GPU at different batch sizes. We compare our Sinkhorn-based matching procedure with Hungarian matching, shown in gray.

| Batch size | 4 | 8 | 16 | 32 | 64 |
|---|---|---|---|---|---|
| Encoder | $8.2 \times 10^{-3}$ | $9.1 \times 10^{-3}$ | $1.4 \times 10^{-2}$ | $2.0 \times 10^{-2}$ | $3.2 \times 10^{-2}$ |
| Decoder | $4.6 \times 10^{-3}$ | $6.8 \times 10^{-3}$ | $8.8 \times 10^{-3}$ | $1.2 \times 10^{-2}$ | $1.5 \times 10^{-2}$ |
| Mask encoding | $7.9 \times 10^{-3}$ | $8.9 \times 10^{-3}$ | $1.1 \times 10^{-2}$ | $1.8 \times 10^{-2}$ | $1.7 \times 10^{-2}$ |
| SK clustering | $3.4 \times 10^{-4}$ | $3.4 \times 10^{-4}$ | $3.4 \times 10^{-4}$ | $3.7 \times 10^{-4}$ | $3.9 \times 10^{-4}$ |
| Sinkhorn matching | $1.4 \times 10^{-3}$ | $1.4 \times 10^{-3}$ | $1.4 \times 10^{-3}$ | $1.4 \times 10^{-3}$ | $1.2 \times 10^{-3}$ |
| Hungarian matching | $5.7 \times 10^{-3}$ | $1.7 \times 10^{-2}$ | $4.0 \times 10^{-2}$ | $9.0 \times 10^{-2}$ | $1.8 \times 10^{-1}$ |

