# OpenReview forum: "Learning Sparse Visual Representations via Spatial-Semantic Factorization"
_ICML.cc/2026/Conference — ICML 2026 regular_

### Official Review · Reviewer_phmJ · 2026-03-08

**Soundness:** 3
**Presentation:** 3
**Significance:** 3
**Originality:** 3
**Overall Recommendation:** 4
**Confidence:** 2

**Summary:**

This paper proposes STELLAR, with the core idea of factorizing visual features into two parts: one part represents the “What” (semantic concepts), and the other represents the “Where” (spatial positions). The authors argue that, in traditional self-supervised learning methods for vision, semantic understanding and image reconstruction often stand in conflict, and that this factorized form can alleviate this tension. The experimental results show that STELLAR can achieve high-quality reconstruction and strong semantic performance with as few as 16 sparse tokens, and performs well on classification, segmentation, and reconstruction tasks. Overall, this is a method paper that attempts to bridge high-level semantic understanding and low-level reconstruction, with a clear motivation and a certain degree of novelty.

**Compliance With Llm Reviewing Policy:**

Affirmed.

**Final Justification:**

The authors have provided helpful clarifications that sufficiently addressed my concerns. The response improves my confidence in the paper, and I will keep my current score.

**Key Questions For Authors:**

1.The paper attributes the main contribution to spatial-semantic factorization and argues that it can alleviate the Invariance Paradox. However, from the ablation study, it can be seen that once Vision Concept Clustering and Set Concepts Alignment are removed, the performance drops significantly. So, how much of the final effect comes from the factorized representation itself, and how much comes from these other objectives? The clarification of this issue will affect the soundness of the paper.

2.The paper sets r = 16 as the default setting and refers to it as a sweet spot where the model enjoys both rich semantics and high-quality reconstruction. Is there any theoretical or empirical basis for this sweet spot? Is it stable under different backbones / foundational priors, datasets, and downstream tasks? Do we need to set r separately for different scenarios? The clarification of this issue will affect the significance of the paper.

**Limitations:**

yes

**Strengths And Weaknesses:**

Soundness:
The paper is technically sound overall, with the motivation of formalizing the conflict between semantics and reconstruction as the Invariance Paradox supported by both deduction and empirical evidence. The experiments are also designed in a detailed and reasonable manner, and the discussion and acknowledgment of limitations are relatively honest.

Presentation:
The paper is written clearly and is well structured, fully discussing related work and differences, and providing a fairly complete set of training hyperparameters for reproducibility. However, the description of the overall framework remains relatively abstract, and it is recommended to add an overview figure of the STELLAR framework.

Significance:
The paper addresses the longstanding conflict between high-level semantics and high-fidelity reconstruction in visual self-supervised learning, and this problem is broadly relevant to the field, helping promote understanding and future research in this area.

Originality:
The paper proposes a new explanatory framework based on the Invariance Paradox, targeting the longstanding tension between reconstruction and semantics in visual self-supervised learning. It introduces a new framework and training scheme based on spatial-semantic factorization, although it does not propose a fundamentally new theory or learning algorithm. Still, it demonstrates sufficient originality and novelty.

---

> ### Author Rebuttal · Authors · 2026-03-30
>
> We thank the reviewer for the positive and thoughtful assessment. We especially appreciate the recognition that the paper addresses an important long-standing tension in visual SSL, and that the motivation, experiments, and overall framework are clear and technically sound. Below we address the main questions directly.
>
> **1. How much comes from the factorized representation itself vs. the additional objectives?**
>
> This is an important question. Our view is that the contribution should be understood as the **factorized latent space together with the SSL objectives that make this latent useful**, rather than factorization alone in isolation.
>
> The factorized representation provides the essential **structural basis/mathematical form**: semantic concepts are represented by sparse tokens $S$, while localization is represented by $L$, so the model can explicitly separate “what” from “where.” This form creates the representational interface in which concept alignment and image reconstruction can co-exist, which is central to addressing the semantics-reconstruction trade-off.
>
> From a mathematical perspective, Eq. (2) partitions variation into semantic and spatial components, so that one part can be driven toward invariance while the other absorbs equivariant variation. Put differently, the factorization and the objectives play different but complementary roles. The factorization defines the space in which semantic/spatial separation is possible at all, while the objectives determine whether that space is actually populated with stable, cross-view concepts rather than degenerate or collapsed tokens.
>
> At the same time, this form alone does *not* guarantee that the sparse tokens become semantically meaningful concepts. The additional objectives, especially Vision Concept Clustering and Set Concepts Alignment, are precisely what drive a clean equivariant partitioning, as empirically illustrated in Fig. 2(a), and make the factorized latent stable and transferable across views.
>
> This interpretation is also supported by Table 3: removing these objectives leads to substantial degradation, and removing Set Concepts Alignment causes particularly severe collapse. So our intended claim is not that factorization alone explains all gains, but rather that the factorized representation is the core design principle, while clustering/alignment are the key learning signals that make this representation identifiable and useful for SSL. We will clarify this more explicitly in the revision.
>
> **2. Why is $r=16$ the sweet spot? Is there a basis for it, and is it stable?**
>
> There are both mathematical intuition and empirical evidence behind this choice.
>
> Mathematically, $r$ is the upper bound on the rank of the visual representation. Intuitively, it controls how many concepts are available to explain a visual scene for both semantic understanding and reconstruction. More complex scenes may require more concepts, while simpler scenes can be represented with fewer.
>
> Importantly, the model is not forced to use all $r$ tokens equally. The representation can effectively use only a subset of the sparse tokens for a given image by assigning very low probability mass to others. We measured the **effective token usage** through the entropy of the learned token distribution, and on the ImageNet-1K validation set it ranges from roughly **11 to 16**. This suggests that the model already learns to use fewer active concepts for simpler scenes, even when the maximum budget is capped at 16.
>
> Empirically, Fig. 2(c) shows that $r=16$ provides the best balance between semantic transfer and reconstruction quality in our main ImageNet-1K setting. Smaller $r$ constrains reconstruction more strongly, while larger $r$ improves reconstruction capacity but weakens the semantic bottleneck. In this sense, $r$ should be viewed not as a brittle hyperparameter, but as a controllable rank budget trading off reconstruction capacity against semantic bottleneck strength. At the same time, performance remains fairly **stable across a reasonably broad range**, so $r=16$ should be viewed as an *empirical default sweet spot* for the backbones and tasks studied here, rather than a universally optimal theoretical constant.
>
> We therefore agree that different scenarios may prefer somewhat different $r$, especially under different model scales, datasets, or downstream emphases. However, the model is also robust across a fairly broad range of $r$ values. We will clarify this discussion in the paper.
>
> **3. Overview figure**
>
> We appreciate the suggestion regarding the framework description. We agree that the method would benefit from an earlier overview figure, and we will move the current architecture figure from the appendix into the main text in the camera-ready version.
>
> We thank the reviewer again for the positive and constructive comments. We believe these clarifications further strengthen the main message of the paper, and we hope they are helpful in reassessing the submission.

---

> > ### Author Rebuttal · Reviewer_phmJ · 2026-04-03
> >
> > Thank you for the rebuttal. The authors have provided helpful clarifications that sufficiently addressed my concerns. The response improves my confidence in the paper, and I will keep my current score.

---

### Official Review · Reviewer_gAmE · 2026-03-08

**Soundness:** 3
**Presentation:** 3
**Significance:** 3
**Originality:** 3
**Overall Recommendation:** 4
**Confidence:** 4

**Summary:**

This paper proposes STELLAR, a self-supervised learning framework that factorizes visual representations into a semantic matrix $\mathbf{S}$ (sparse concept tokens, the "What") and a localization matrix $\mathbf{L}$ (spatial distribution, the "Where"), with $Z = LS$. The motivation is the "Invariance Paradox"—the tension between transformation-invariant features needed for semantics (DINO-style) and transformation-equivariant features needed for reconstruction (MAE-style). By isolating invariance in $\mathbf{S}$ and equivariance in $\mathbf{L}$, the framework can simultaneously support augmentation alignment and pixel-level reconstruction. Training combines reconstruction loss, prototype clustering with Sinkhorn-Knopp balancing, OT-based set alignment, CLS alignment, and KoLeo regularization. Experiments on ImageNet-1K show that 16 sparse tokens achieve 79.10% linear probing accuracy (ViT-H) and 2.60 FID, with competitive segmentation on ADE20K and Pascal VOC.

**Compliance With Llm Reviewing Policy:**

Affirmed.

**Final Justification:**

I thank the authors for their detailed response and for addressing my questions. Their clarifications were helpful, and I will maintain my current score.

**Key Questions For Authors:**

1. **How does STELLAR perform from scratch with matched total compute?** The random-prior result (65.28%) is far below the MAE-initialized result (73.26%). A comparison matching total FLOPs (pretraining + STELLAR) would clarify how much value the factorization adds on its own. A strong result here could raise my assessment.

2. **What is the relationship to slot attention / object-centric models?** Both decompose images into small sets of tokens with spatial attention maps. Have you compared against slot attention baselines, and does the factorization offer advantages on multi-object scenes (e.g., COCO, MOVi)? This would strengthen positioning.

3. **How sensitive is the method to prototype count $K$?** You use $K = 16384$, which is very large relative to $r = 16$ tokens. What happens with $K = 1024$ or $K = 4096$?

4. **Can you provide per-image reconstruction metrics (PSNR, SSIM)?** FID can be misleading for reconstruction quality. Per-image metrics would better characterize whether the 16-token bottleneck preserves fidelity or produces plausible but unfaithful outputs.

5. **Why does linear probing accuracy decrease as $r$ increases (Fig. 2c)?** More tokens should provide at least as much information. Is this a mean-pooling artifact? Have you tried attention-pooling or max-pooling?

**Limitations:**

The authors acknowledge underperformance on object-centric datasets (Pets, Food) due to mean-pooling dilution, which is fair. However, several important limitations are not discussed: (1) dependence on pretrained backbones—the performance gap between pretrained and random init is significant but unacknowledged; (2) computational overhead relative to simpler baselines (MAE, DINO) given the multiple loss terms, OT solving, and Sinkhorn clustering; (3) scalability beyond 224×224 resolution; (4) the societal impact discussion is minimal.

**Strengths And Weaknesses:**

**Strengths:**

- The spatial-semantic factorization ($Z = LS$) is mathematically clean and a genuinely novel contribution as a core SSL latent space. The equivariant partitioning analysis (Fig. 2a) convincingly shows that $\mathbf{S}$ remains stable under spatial shifts while $\mathbf{L}$ absorbs the variation—this is strong empirical evidence that the factorization works as intended.
- The ablation (Table 3) is thorough and clearly demonstrates necessity of each component, especially the dramatic collapse without set alignment (row C).
- The demonstration that only 16 tokens can support both high-quality reconstruction and competitive semantics is a practically significant finding for efficient visual tokenization, particularly relevant for LLM integration.
- Bridging discriminative and generative SSL is an important and timely problem. The factorized representation is a conceptual contribution likely to influence future work in multimodal systems and interpretable vision models.
- The combination of OT-based set alignment for unordered sparse tokens with prototype clustering is a creative solution to a genuine technical challenge.

**Weaknesses:**

- The "Invariance Paradox" argument (Section 3), while intuitive, is presented loosely. The lower bound on $\|\partial Z / \partial \theta\|_F$ relies on unspecified conditions regarding the decoder Jacobian and uses "$\gtrsim$" without clear assumptions. This is a heuristic argument rather than a rigorous impossibility result, and should be stated as such.
- The headline ViT-H results (79.10%, 2.60 FID) use a pretrained MAE backbone, making it hard to separate STELLAR's contribution from the foundation model. The random-prior results (Table 4: 65.28%) are substantially weaker—closer to MAE's own baseline. How much comes from the factorization vs. the pretrained features?
- The ViT-B linear probing result (73.26%) still trails DINO (76.46%) and iBOT (76.40%) by ~3 points. The abstract's claim of "matching semantic performance of dense backbones" is somewhat overstated at base scale.
- Reconstruction is evaluated via FID and LPIPS, but FID is distributional—it does not assess per-image fidelity. Per-image metrics (PSNR/SSIM) would be more informative for evaluating whether the 16-token bottleneck preserves spatial fidelity or just produces plausible outputs.
- Many individual components are borrowed from existing work (Sinkhorn-Knopp from SwAV/DINOv2, learnable queries from DETR/Perceiver, KoLeo regularization, MaskGIT-VQGAN decoder). The novelty lies in their combination and the factorization principle.
- The relationship to slot attention (Locatello et al., 2020) is not discussed despite significant conceptual overlap—both decompose scenes into latent slots with spatial attention maps. A discussion of this connection seems important.
- The paper lacks a clear architectural diagram until the appendix (Fig. 4). Bringing it into the main text would improve readability. Some important details (momentum encoder for random-prior training, masking strategy, decoder architecture) are also deferred.

---

> ### Author Rebuttal · Authors · 2026-03-30
>
> We thank the reviewer for the careful and insightful feedback. We especially appreciate the recognition of the spatial-semantic factorization as the core contribution and the positive assessment of the equivariant partitioning evidence. We address the main points below and will incorporate these clarifications into the revision.
>
> **1. Invariance paradox**
>
> We agree that Sec. 3 should be stated more carefully. Our intention was to provide a *heuristic tension analysis*, not a formal impossibility theorem. We will revise the wording accordingly and clarify that the main contribution is the factorized representation $Z(X)=L(X)S(X)$ and its empirical behavior, rather than a theorem-level claim.
>
> **2. Semantic performance.**
>
> We agree that STELLAR trails DINO and iBOT on IN1K linear probing at ViT-B scale, and we will soften the abstract claim of “matching semantic performance of dense backbones.” Our intended claim is instead that STELLAR achieves a stronger **semantics-reconstruction trade-off** with a compact factorized latent. Appendix Table 5 further supports this distinction:
>
> | Method  |   CLS | Dense | Factorized |
> | :---- | ----: | ----: | ---: |
> | DINO | 76.46 | 70.31 | - |
> | MAE  | 65.61 | 66.32 | - |
> | iBOT | 76.40 | 71.44 | - |
> | STELLAR |-| 72.21 | 73.26 |
>
> DINO/iBOT obtain their strongest semantics from the CLS token, while their reconstruction-feasible dense features are weaker on that semantic metric. In contrast, STELLAR’s factorized latent remains both semantically strong and reconstruction-feasible.
>
> **3. PSNR and SSIM.** Following the reviewer’s suggestion, we added per-image metrics in addition to LPIPS/FID in Table 1:
>
> | Model | # Tokens | FID ↓ | LPIPS ↓ | PSNR| SSIM|
> |:-|-:|-:|-:|-:|-:|
> | MAE  | 196 |  3.02 |  0.208 |16.54 |0.398|
> | DINO | 196 |  3.27 |  0.213 |  16.36 |  0.390 |
> | TiTok |   64 |  1.99 |  0.257 |  16.01 |  0.372 |
> | TiTok |  32 |  2.75 |  0.328 |  14.97 |  0.331 |
> | STELLAR  | 16 |  3.06 |  0.208 |  16.57 |  0.397 |
> | STELLAR  | 196 |  2.85 |  0.209 |  16.58 |  0.399 |
> | STELLAR(H) | 16 |  2.60 |  0.173 |  17.91 |  0.468 |
>
> These results support the same conclusion as Table 1: STELLAR’s factorized latent is competitive with or slightly better than MAE/DINO in **per-image fidelity**, while lower FID alone does not necessarily imply better reconstruction faithfulness (e.g. TiTok).
>
> **4. Accuracy v.s. $r$**
>
> We tested whether this is only a mean-pooling artifact. The same pattern holds across pooling strategies. So the drop at larger $r$ is not only a pooling artifact, but reflects a real trade-off between stronger bottlenecked semantics and reconstruction capacity.
>
> | $r$ | attn |  mean |   max |
> | :-- | --: | --: | --: |
> | 8   | 72.86 | 72.97 | 72.33 |
> | 16  |73.05 | 73.26 | 72.17 |
> | 24  | 72.15 | 72.17 | 70.49 |
>
> **5. Prototype sensitivity.**
>
> We also ran the requested $K$-sensitivity study. Performance is stable from $K=4096$ upward, while $K=1024$ performs worse and exhibits less stable clustering/alignment. This suggests that a sufficiently large $K$ is important as a fine-grained visual concept vocabulary.
>
> | $K$   | IN1K ↑ | ADE20K ↑ |
> | :-- | --: | --: |
> | 1024|70.79 |30.21|
> | 4096|73.12 | 30.64|
> | 8192|73.31 | 30.98|
> | 16384|73.26 | 31.33|
>
> **6. Pretraining with matched compute**
>
> We agree that the role of the pretrained backbone should be made more explicit. Table 4 shows that random-init STELLAR reaches 65.28 on IN1K, already close to MAE’s 66.32, while MAE-init improves performance substantially to 73.26. We also estimated the total budget of the two setups: random-init STELLAR and MAE-pretrain + STELLAR have similar total cost (41.48h vs. 41.24h; random-init uses 1.086× FLOPs). Thus, the gain is not simply due to a larger compute budget. Our interpretation is that STELLAR from scratch already learns MAE-level semantics, while a strong prior and the STELLAR objective are complementary and together yield the best performance. We will enhance this discussion.
>
> **7. Related work**
>
> We agree this connection should be discussed more explicitly. Conceptually, both STELLAR and slot attention use a small set of tokens. However, the intended regime is different: STELLAR is a general self-supervised representation learner that jointly supports semantic transfer and reconstruction. We will expand this discussion in the related-work section.
>
> **8. Use of standard components**
>
> This was intentional. We deliberately used simple, standard components so that the experiments isolate the contribution of the **representation form and SSL objective**, rather than introducing confounds from architectural novelty. We will make this positioning clearer. We will also move the appendix architecture figure into the main text in the final version.
>
> We thank the reviewer again for the thoughtful comments. The requested experiments were very helpful in sharpening both the interpretation and the claims, and we hope these clarifications are helpful in reassessing the paper.

---

> > ### Author Rebuttal · Reviewer_gAmE · 2026-04-01
> >
> > I thank the authors for their detailed response and for addressing my questions. Their clarifications were helpful, and I will maintain my current score.

---

### Official Review · Reviewer_J3G4 · 2026-03-12

**Soundness:** 2
**Presentation:** 3
**Significance:** 3
**Originality:** 3
**Overall Recommendation:** 4
**Confidence:** 4

**Summary:**

The paper introduces STELLAR, a self-supervised learning framework designed to reconcile the conflict between semantic invariance and spatial equivariance in visual representations. It achieves this by factorizing the latent space into a low-rank product of sparse semantic concept tokens and a spatial localization matrix ($Z=LS$). The model is trained using a combination of low-rank image reconstruction, optimal transport-based set alignment, and prototype clustering. Empirical results on ImageNet-1K demonstrate that using as few as 16 tokens, STELLAR achieves a strong balance of high-fidelity reconstruction and robust semantic transfer for downstream tasks.

**Compliance With Llm Reviewing Policy:**

Affirmed.

**Final Justification:**

I thank the authors for their detailed and thoughtful rebuttal. They have successfully addressed my technical queries.

Overall, I think this is a solid paper with valid contributions, and I lean towards accepting it, but I believe a Borderline Accept is the most appropriate rating.

**Key Questions For Authors:**

How exactly are the reconstruction metrics computed across models with fundamentally different decoder objectives (e.g., VQGAN token prediction versus direct pixel reconstruction)? Providing details on the "shared decoder" inputs and confirming strict parity in loss functions and capacities would clarify the soundness of these comparisons and strengthen the evaluation.

Since the localization matrix relies on the dense feature map, what are the actual real-world inference compute and memory savings? A wall-clock latency, memory, and FLOPs comparison against baselines like MAE or DINO would substantiate the practical significance of the efficiency claims beyond mere representational compression.

Can dense-prediction tasks, such as segmentation, be performed directly from the factorized representation without accessing the dense feature map? Comparing direct factorized segmentation to dense feature segmentation would isolate the specific utility of the proposed sparse representation and clarify the source of the transfer strength.

How does the STELLAR framework compare to other token-reduction or object-centric learning approaches (like Slot Attention) when adapted to a similar self-supervised setting? Including this discussion would greatly strengthen the paper's originality and positioning within the broader literature.

**Limitations:**

While the paper includes a brief impact statement , it does not adequately discuss the specific limitations and potential negative societal impacts of this work. I recommend adding a section that critically examines the heavy compute footprint required for training (e.g., 16 NVIDIA A100 GPUs, large prototype banks) and how that impacts accessibility for researchers with fewer resources. Additionally, given the model's strong high-fidelity reconstruction capabilities, a brief, thoughtful discussion on the dual-use nature of such representations in sensitive domains like surveillance would align better with ICML guidelines.

**Strengths And Weaknesses:**

The paper addresses a highly relevant and open problem in self-supervised learning by proposing a clean and compelling theoretical framing of the invariance-equivariance tension. Originality shines through the cohesive integration of low-rank reconstruction, prototype-based clustering, and a principled optimal transport approach for set concept alignment. The presentation is a notable strength; the problem is articulated clearly with intuitive diagrams, and the training protocols are thoroughly detailed. Furthermore, the experimental validation is rigorous, offering a broad suite of evaluations across reconstruction, global classification, and semantic segmentation, alongside decisive ablations that justify the method's components. This approach shows significant promise in shaping dense backbones into more region-aware features, potentially benefiting multimodal interfaces.However, the submission has several limitations regarding soundness and positioning. The theoretical derivation of the "Invariance Paradox" is presented somewhat informally and requires stricter mathematical justification to support its claims. Methodologically, calculating the localization matrix relies on computing the dense feature map, which means the computational efficiency gains during inference are primarily in representational storage rather than runtime compute, an aspect that is under-discussed. Experimentally, the reconstruction baseline comparisons lack strict parity due to varying decoder choices and objectives across models, and it remains unclear whether the strong segmentation transfer stems from the sparse factorization itself or general backbone improvements, as dense features are utilized for these tasks. Finally, the paper would benefit from a more thorough contextualization within closely related work, specifically object-centric slot-based representations and token-reduction methods.

---

> ### Author Rebuttal · Authors · 2026-03-31
>
> We thank the reviewer for the careful and thoughtful feedback. We especially appreciate the recognition of the importance of the problem, the originality of the factorized latent design, and the strength of the experimental evaluation. We address the main concerns below and will incorporate these clarifications into the revision.
>
> **1. Clarifying Invariance Paradox**
>
> We agree that the current presentation of the “Invariance Paradox” is more heuristic than theorem-like, and we will revise the wording accordingly. Our intended claim is *not* a general impossibility result. Rather, the point is that when a *single latent* is optimized to be simultaneously invariant for semantic alignment and sufficiently equivariant/informative for reconstruction, these objectives place competing pressures on the same representational channel unless semantic and spatial variation can be separated.
>
> STELLAR addresses this through the factorized structure $Z = LS$, where the semantic tokens and spatial assignments play different roles. Our support for this claim is therefore primarily empirical: the equivariant partitioning behavior in Fig. 2(a), the semantics-reconstruction trade-off in the main results, and the ablations showing the contribution of the factorized design. We will revise the wording to make this interpretation more precise.
>
> **2. Reconstruction protocol**
>
> Thank you for raising this point. In Table 1, reconstructions for MAE and DINO are obtained by training the same STELLAR decoder with the same loss on top of the *frozen features* from each encoder. In other words, the comparison is intentionally decoder-controlled, so that reconstruction metrics reflect the information content of the representation rather than differences in decoder architecture. The only exception is TiTok, whose non-factorized sparse latent requires a dedicated decoder architecture. We additionally reported its native larger decoder in the paper. We will make this protocol more explicit in the revision.
>
> **3. Compute efficiency**
>
> We agree with the reviewer that as the localization matrix is computed from the dense feature map, the main practical benefit is *not* lower encoder-side compute. We benchmarked MAE and STELLAR more carefully using batch size 128 on a single A100:
>
> | Model   | Wall-clock (ms) ↓ | FLOPs | Peak Mem (GB) ↓ |
> | :------ | ----: | ----: | ----: |
> | MAE     |  40.2 |  1.0× |  5.16 |
> | STELLAR | 43.7 |  1.1× | 5.71 |
>
> Thus, the cost of representation extraction remains close to standard ViT models such as MAE. We will revise the paper to clarify that the practical efficiency advantage is primarily the **compact sparse interface for downstream use**, rather than cheaper encoder-side inference. This compactness becomes especially relevant when visual tokens are passed to a downstream transformer/MLLM: replacing a dense grid (196 tokens) with 16 sparse tokens reduces token count by $12.25\times$, which in turn reduces quadratic attention cost in downstream processing by roughly $150\times$
>
> **4. Factorized segmentation**
>
> This is an important question, and we trained segmentation probes on the low-rank representation without using the dense feature map. The factorized representation alone retains non-trivial spatial signal despite its much smaller effective per-token dimension. However, there remains a substantial gap relative to the dense map. We therefore confirm that the current dense-task results should be interpreted as "sparse factorized training improves the resulting dense backbone features", rather than that the low-rank representation already fully replaces dense maps for segmentation. We will further clarify this distinction in the paper.
>
> | Rep. | Dim/patch | ADE20K |   VOC | City |
> | :-----| ---: | -----: | ----: | ----: |
> | Low-rank    |    79 |  17.58 | 70.00 |13.60 |
> | Dense   |   768 |  31.33 | 81.83 |  27.74 |
>
> **5. Related work**
>
> We agree that this discussion should be strengthened. STELLAR shares with slot methods the idea of using a small set of tokens, but designed as a general self-supervised representation learner for jointly supporting semantic transfer and reconstruction. Likewise, unlike token-reduction methods that primarily compress or prune tokens, STELLAR is trained to preserve both reconstructive and semantic utility through the factorized latent. We will expand this discussion in the related-work section.
>
> **6. Limitations**
>
> We agree that this section should be stronger with discussion of: (i) training compute footprint and the resulting accessibility implications, (ii) the efficiency gains are mainly in representation compression, and (iii) the dual-use risk of stronger high-fidelity reconstruction, including potential misuse in sensitive applications.
>
> We thank the reviewer again for the constructive comments. These clarifications and additional experiments were very helpful in sharpening both the claims and the interpretation, and we hope they are helpful in reassessing the paper.

---

> > ### Author Rebuttal · Reviewer_J3G4 · 2026-04-01
> >
> > I thank the authors for their response and for addressing my questions. My main concerns were addressed.

---

### Official Review · Reviewer_jSYr · 2026-03-15

**Soundness:** 2
**Presentation:** 2
**Significance:** 2
**Originality:** 3
**Overall Recommendation:** 3
**Confidence:** 4

**Summary:**

The authors propose to bridge the gap between SSL methods that apply the loss on a global vector and ones that apply a loss patch-wise, often with reconstruction. To do so, they propose to factorize the representations to apply an invariance loss at a semantic level, and keep a global structure for the reconstruction objective. A pretrained model is then finetuned with this objective, leading to improved performance on recognition and reconstruction tasks.

**Compliance With Llm Reviewing Policy:**

Affirmed.

**Final Justification:**

After the author's rebuttal, my main concern regarding the use of pretrained encoders remains. Quoting the authors "we view STELLAR as a self-supervised representation learning framework rather than a task-specific finetuning procedure", however most results are presented when finetuning a pretrained network. Results from scratch are less convincing, with relatively low performance on image classification for example, restricting the use of the method.

This leads to a work that I feel is ambiguous and unclear in the contributions, making me lean towards rejection.

**Key Questions For Authors:**

For the main points, refer to weaknesses.

Regarding DINO, how are reconstructions obtained to measure LPIPS ? The method does not support a decoder out of the box so was one trained for it ?

**Limitations:**

Yes

**Strengths And Weaknesses:**

**Strengths**
- The idea to use sparse representations is interesting as it avoids the need to obtain a singular global representation for the invariance loss.
- The method shows competitive performance with baselines across reconstruction, segmentation and classification tasks. The method performs particularly well on segmentation tasks
- Ablations are thorough, helping understand the method's behavior more precisely.

**Weaknesses**
- Some claims about spatial information in existing works are a bit misleading. Lines 16-17 ”inherently discards the spatial coordinates”, Lines 40-42 “the model is pressured to discard spatial variance”. While this may be true of the CLS token, methods such as DINO work remarkably well for tasks such as image segmentation when using the patch embeddings. This is however a byproduct of the architecture as no direct loss is applied on the patch embeddings (in DINO at least, there is one in iBoT/DINOv2 for example), but this should be made clearer.
- Related work on methods combining reconstruction and a contrastive object should be discussed, e.g. CMAE[1] or CAN[2]. More different, but SODA[3] also tackles both reconstruction and semantic tasks and would probably deserve a mention/inclusion in the results.
- On representation learning tasks, the method generally underperforms other methods at similar scale, e.g. DINO ViT-B or iBoT Vit-B/L
- The method seems to mainly work when using an already pretrained encoder as a starting point and finetuning it. This point is mentioned as the use of a “foundation prior”, and from my understanding the only experiment from scratch is the last line of Table 4, which achieved underwhelming results. The method still seems effective at improving MAE and provides a strong end model, but this is an important limitation. It should also be made clearer in the main text, rather than being clearly described only in the appendix.

**Minor points:**
- Typo line 243 "tsansport"
- Table 2: The Method column perhaps too detailed which makes it more confusing than really highlighting differences. For example, what is the difference between Rec (Since STELLAR predicts MaskGIT-VQGAN tokens, line 261) and Tok MIM ? The titles for each subsection of the table seem sufficient.
- For Table 2, were all numbers obtained by the authors ? Or are some taken from the original works ?
- Including figures 4 and 5 in the main manuscript would help the presentation and make the method's behavior clearer.
- Figure 3, a validation/test set should be used, not the train one.


[1] Huang, Zhicheng, et al. "Contrastive masked autoencoders are stronger vision learners." IEEE Transactions on Pattern Analysis and Machine Intelligence 46.4 (2023): 2506-2517.

[2] Mishra, Shlok, et al. "A simple, efficient and scalable contrastive masked autoencoder for learning visual representations." arXiv preprint arXiv:2210.16870 (2022).

[3] Hudson, Drew A., et al. "Soda: Bottleneck diffusion models for representation learning." Proceedings of the IEEE/CVF Conference on Computer Vision and Pattern Recognition. 2024.

---

> ### Author Rebuttal · Authors · 2026-03-30
>
> We thank the reviewer for the careful reading and constructive feedback. We are encouraged that the reviewer found the sparse factorized representation interesting, the overall empirical performance competitive, and the ablations helpful. We address the main concerns below and will incorporate these clarifications in the revision.
>
> **1. Spatial-information wording**
>
> We agree that our current wording is too broad, especially phrases such as “inherently discards spatial coordinates” and “the model is pressured to discard spatial variance.” Our intended claim is not that methods such as DINO lack useful spatial structure in their patch embeddings, nor that they cannot support dense tasks. Rather, our point is that the representations that are strongest for global semantic alignment are not necessarily the same ones that are most suitable for spatially faithful reconstruction.
>
> This distinction is reflected in Appendix Table 5 by linear probing different features:
>
> |   |   CLS | Dense | Factorized (ours) |
> | :------ | ----: | ----: | ---------: |
> | DINO    | 76.46 | 70.31 |          – |
> | MAE     | 65.61 | 66.32 |          – |
> | iBOT    | 76.40 | 71.44 |          – |
> | STELLAR |     – | 72.21 |      73.26 |
>
> These results suggest that DINO/iBOT preserve useful dense structure, while their strongest semantics are concentrated in the global token. STELLAR instead aims to learn a compact factorized latent that better balances semantic utility and reconstruction utility within a single representation. We will revise the corresponding statements in the abstract and main text accordingly.
>
> **2. Semantic performance**
>
> We agree that STELLAR generally underperforms the strongest JE baselines on pure global semantics at similar scale, especially at ViT-B. We will soften the current wording. Our intended claim is not that STELLAR uniformly outperforms DINO/iBOT on classification, but that it achieves a stronger semantics–reconstruction trade-off through a sparse factorized latent that can support both discriminative and generative uses.
>
> **3. Dependence on pretrained encoders**
>
> We agree this is an important limitation and should be stated more clearly in the main text. Our claim is not that STELLAR replaces large-scale pretraining; rather, it is most effective as an objective for reshaping a strong prior into a sparse holistic representation. At the same time, Table 4 suggests the gains are not solely due to the prior: from random initialization, STELLAR reaches 65.28 on IN1K, close to MAE’s 66.32, while also providing comparable reconstruction quality; with a stronger MAE prior, performance improves substantially to 73.26. We will emphasize this limitation and interpretation in the main text.
>
> **4. Reconstruction protocol**
>
> Thank you for flagging this point. For the reconstruction comparisons in Sec. 5.2 / Table 1, reconstructions for DINO, MAE, and other baselines are obtained using a decoder-controlled protocol: we freeze the baseline encoder, train the same STELLAR decoder on top of its frozen features, and evaluate FID/LPIPS using the same setup. Thus, DINO is not evaluated with a native decoder, but with the same reconstruction head trained on top of frozen DINO features. This was done so that FID/LPIPS reflect the information content of the learned representation rather than differences in decoder capacity. TiTok is the only exception, whose non-factorized sparse latent requires dedicated decoder architecture, which we reported results with its native larger decoder in the paper. We will make this protocol explicit in the revision.
>
> **5. Related work**
>
> We agree that related hybrid methods should be discussed more explicitly. We will revise the related-work section to cover CMAE and CAN as reconstruction-plus-contrastive hybrids, and SODA as another approach aimed at supporting both semantic and generative behavior through a bottlenecked representation. Our intended distinction is that STELLAR emphasizes sparse factorization as the mechanism for improving the balance between these objectives.
>
> **6. Baseline details**
>
> We confirm that all numbers in Table 2 were obtained from our own evaluation under a consistent setup using the official released checkpoints for each baseline. We will state this more clearly in the paper. We also agree that the current “Method” column is overly detailed and will simplify the notation to improve readability.
>
> **7. Presentation fixes**
>
> We will fix the typo on line 243, clarify the Table 2 notation, move Figures 4 and 5 into the main paper if space permits, and improve Figure 3 with a clear train/validation split definition.
>
> We thank the reviewer again for the constructive comments. We believe these clarifications improve the paper materially by narrowing the claims, making the reconstruction protocol explicit, and being more transparent about the role of pretrained priors. We hope these clarifications are helpful in reassessing the paper.

---

> > ### Author Rebuttal · Reviewer_jSYr · 2026-04-01
> >
> > Thank you for your answers.
> > A lot of my questions pertaining to presentation are well addressed.
> > However some of my key concerns remain, and may not be easily addressable with an unsupervised manuscript revision or in the time of the rebuttal. The "foundation prior" aspect remains for me the biggest limitation, even though the authors do demonstrate that finetuning an MAE model with the STELLAR objective improves performance.
> >
> > I think that the method has potential, not used standalone but perhaps as an additional regularisation rather than finetuning, but that the changes required to the work may be beyond the scope of a simple rebuttal.
> > I will keep my score as is for now.

---

> > > ### Author Response · Authors · 2026-04-04
> > >
> > > Thank you again for the thoughtful engagement, and we appreciate your recognition of the paper’s potential. We also appreciate your remaining concern regarding how the method should be viewed when the strongest regime is built on a strong prior.
> > >
> > > Our view is that the remaining difference is primarily about how to position the contribution. In particular, we view STELLAR as a self-supervised representation learning framework rather than a task-specific finetuning procedure, since the training remains fully self-supervised and is aimed at learning a more general-purpose, unified representation rather than adapting to a downstream task.
> > >
> > > More specifically, STELLAR introduces a different self-supervised representation family, namely a factorized latent $Z=LS$ with explicit separation between semantic concepts and localization. The associated objectives are designed to make this latent jointly support semantic utility and reconstruction within a unified SSL framework.
> > >
> > > We will make this intended positioning clearer in the revision, and hope this clarification is helpful in understanding the intended contribution and scope of the work.

---

### Decision · Program_Chairs · 2026-04-30

**Decision:**

Accept (regular)

**Comment:**

The paper proposes a factorized latent representation that models images as a product of a sparse set of semantic tokens ("what") and a spatial assignment matrix ("where") mapping those tokens to specific locations.  This amounts to a low-rank factorization of the feature map for an image.  Combined with self-supervised learning methods, it enables systems to simultaneously learn semantic invariance (enabling downstream tasks such as classification), and spatial equivariance (enabling reconstruction and generation tasks).  After the rebuttal and discussion, three of four reviewers lean toward accept, stating that the rebuttal addressed their concerns.  Reviewers gAmE and phmJ note the novelty and originality of the approach.

Reviewer jSYr gives a weak reject final rating, with the complaint that the system relies on finetuning pretrained encoders for best results.  The AC does not agree that this is a significant concern; the novelty of the approach does hinge on this result and the rebuttal correctly points out that the system also works when trained from scratch ("Rand" in Table 4).  The AC also disagrees with Reviewer J3G4's comments about limitations; using 16 GPUs does not constitute heavy compute in the era of foundation models.

Overall, this work introduces a novel and promising technique for formulating latent representations for modeling images, along with sufficient experimental proof-of-concept.  The AC believes it clearly meets the bar for acceptance.